# Pixel-level Certified Explanations via Randomized Smoothing

**Alaa Anani** [1][2]   **Tobias Lorenz** [2]   **Mario Fritz** [2]   **Bernt Schiele** [1]

## Abstract

Post-hoc attribution methods aim to explain deep learning predictions by highlighting influential input pixels. However, these explanations are highly non-robust: small, imperceptible input perturbations can drastically alter the attribution map while maintaining the same prediction. This vulnerability undermines their trustworthiness and calls for rigorous robustness guarantees of pixel-level attribution scores. We introduce the first certification framework that guarantees pixel-level robustness for any black-box attribution method using randomized smoothing. By sparsifying and smoothing attribution maps, we reformulate the task as a segmentation problem and certify each pixel's importance against $\ell_2$-bounded perturbations. We further propose three evaluation metrics to assess certified robustness, localization, and faithfulness. An extensive evaluation of 12 attribution methods across 5 ImageNet models shows that our certified attributions are robust, interpretable, and faithful, enabling reliable use in downstream tasks. Our code is at https://github.com/AlaaAnani/certified-attributions.

## 1. Introduction

Computer vision models are highly successful in high-stakes applications, such as autonomous driving (Kaymak & Uçar, 2019; Zhang et al., 2016), medical imaging (Kayalibay et al., 2017; Guo et al., 2019), and the judicial system (Nissan, 2017). In such domains, understanding the reasoning behind model predictions through explainability methods is paramount. Post-hoc attribution methods identify image pixels most influential for a model's prediction. However, attribution methods are highly non-robust: even impercep-

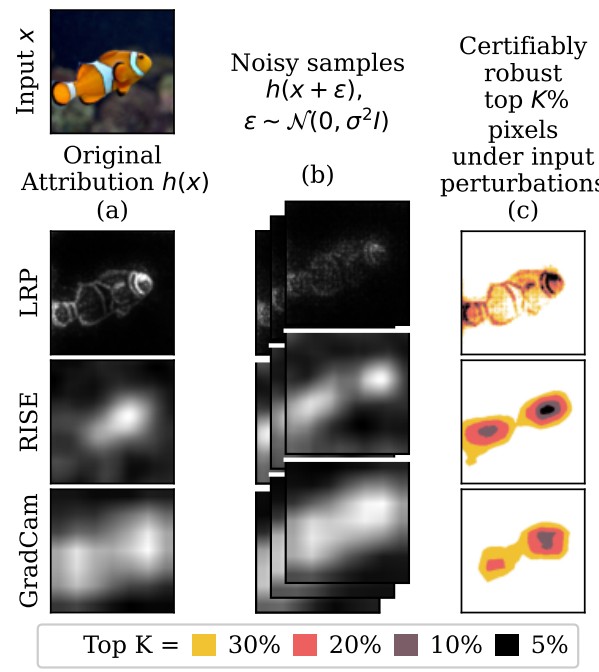

Figure 1: **Pixel-level certified explanations via randomized smoothing.** (a) The original attribution $h(x)$ lacks robustness guarantees and changes under small input perturbations. (b) We sample noisy attributions $h(x + \epsilon)$, and (c) certify which pixels robustly remain in the top-$K\%$. Colored pixels show certifiably robust top $K$ pixels across different attribution methods (LRP, RISE, Grad-CAM).

tible input perturbations can significantly alter attribution maps while leaving predictions unchanged (Ghorbani et al., 2019; Kindermans et al., 2019; Dombrowski et al., 2019; Subramanya et al., 2019; Kuppa & Le-Khac, 2020; Zhang et al., 2020). This vulnerability undermines trust in attribution methods, especially when explanations directly influence critical decisions (Sheu & Pardeshi, 2022; Xi et al., 2023). Consequently, pixel-level certification methods that rigorously guarantee the robustness of each pixel's importance value under input perturbations become crucial for trustworthy model explanations.

Previous work addresses attribution robustness through certifying bounds for image-level metrics with respect to input perturbations — typically measuring the similarity of attri-

[1]Max Planck Institute for Informatics, Saarland Informatics Campus, Saarbrücken, Germany [2]CISPA Helmhotz Center for Information Security, Saarbrücken, Germany. Correspondence to: Alaa Anani <aanani@mpi-inf.mpg.de>.

*Proceedings of the 42nd International Conference on Machine Learning*, Vancouver, Canada. PMLR 267, 2025. Copyright 2025 by the author(s).

butions between perturbed and clean inputs (Wang & Kong, 2023; 2024; Dhurandhar et al., 2024; Levine et al., 2019; Liu et al., 2022; Chen et al., 2022). However, image-level bounds are too coarse, as they (i) fail to capture pixel-level robustness, (ii) lack visual interpretability as they only produce a single number, and (iii) can be disproportionately affected by a small non-robust subset of pixels.

We address this challenge and introduce the first certification method that guarantees pixel-level robustness for any black-box attribution method, producing fine-grained, interpretable certified attributions. Our key insight is to reframe attribution as a segmentation task by sparsifying the continuous attribution maps into binary pixel-importance classes: "1" for the top $K\%$ most important pixels and "0" for the rest. This defines a sparsified attribution function, whose output we certify using Randomized Smoothing (Fischer et al., 2021; Anani et al., 2024), a framework that certifies high-dimensional outputs of segmentation models against $\ell_2$-norm input perturbations. Each pixel is certified as either robustly important ("1"), robustly unimportant ("0"), or abstaining ("$\oslash$") if no guarantee can be given. Figure 1 shows an example of certified attributions for three attribution methods at different sparsification levels $K$.

Leveraging our certification framework, we comprehensively compare 12 different attribution methods spanning backpropagation-based, activation-based, and perturbation-based methods on 5 different ImageNet classifiers, including CNNs (e.g., ResNet-152) and transformer (ViT-B/16) architectures. Leveraging our pixel-wise certificates, we create the first certified attribution maps, highlighting image pixels that are robustly influential for predictions (Figures 1 to 4). For quantitative evaluations, we propose three metrics that measure desirable properties of certified attributions: (i) the percentage of certified pixels (%certified), measuring the attribution method's robustness, (ii) Certified Grid Pointing Game (Certified GridPG), an extension of GridPG (Bohle et al., 2021) that measures robust localization by computing the percentage of correctly localized *and* certified attributions, and (iii) a deletion-based faithfulness score that measures the confidence drop when progressively removing the pixels with top $K\%$ certified attributions. Our experiments show significant differences between methods in all four dimensions, with LRP and RISE demonstrating the best trade-offs between certified robustness, localization, faithfulness, and meaningful visual attributions.

**Contributions** To summarize, our contributions are: (i) We introduce the first pixel-level certification framework for *any* black-box attribution method, providing rigorous robustness guarantees under $\ell_2$-bounded perturbations (Section 4), (ii) We present three certified evaluation metrics: %certified to assess robustness, Certified GridPG for robust localization, and a deletion-based score for evaluating

faithfulness (Section 5), (iii) We create the first certified attribution maps that can be used in downstream tasks (Sections 7.1), (iv) We analyze trade-offs across robustness, localization, and faithfulness of 12 attribution methods on 5 ImageNet models, including CNNs and transformers, showing that LRP and RISE strike the best balance, offering actionable insights into trustworhty and reliable attribution (Sections 7.2, 7.3, 7.4, 7.5).

## 2. Related Work

**Machine learning explainability** Post-hoc attribution methods produce maps that show which pixels in an input image are important to a classifier's output. Attribution methods fall into three categories: (i) **backpropagation-based**, which typically use gradients with respect to the input (Simonyan, 2014; Springenberg et al., 2014; Sundararajan et al., 2017; Shrikumar et al., 2017), (ii) **activation-based**, which weigh activation maps to determine the importance of pixels in relation to the network's final layer (Selvaraju et al., 2017; Chattopadhay et al., 2018; Ramaswamy et al., 2020; Jiang et al., 2021), and (iii) **perturbation-based**, which perturb the input differently (e.g. via occlusions or masking) to observe changes in the network output (Dabkowski & Gal, 2017; Fong & Vedaldi, 2017; Petsiuk et al., 2018b; Ribeiro et al., 2016; Zeiler, 2014).

Gradient (Grad) (Simonyan, 2014), one of the earliest backpropagation-based methods, generates saliency maps by computing input gradients. Guided Backpropagation (GB) (Springenberg et al., 2014) refines this by retaining only positive gradients, while Integrated Gradients (IntGrad) (Sundararajan et al., 2017) accumulates gradients along a path from a reference baseline to reduce gradient noise. Input × Gradient (IxG) (Shrikumar et al., 2017) multiplies the input with its gradient to highlight pixel importance. Layerwise Relevance Propagation (LRP) (Bach et al., 2015), in contrast, redistributes the network's output relevance backward through the layers using a set of propagation rules, ensuring conservation of relevance while highlighting features that contribute most to the model's decision.

Among activation-based methods, Class Activation Mapping (CAM) (Zhou et al., 2016) projects class-specific weights onto the final convolutional feature maps to produce coarse localization. Grad-CAM (Selvaraju et al., 2017) uses gradients and feature maps for coarse class-discriminative localization, while Grad-CAM++ (Chattopadhay et al., 2018) extends it for finer details. Ablation-CAM (Ramaswamy et al., 2020) removes neurons to assess their impact on the input, and Layer-CAM (Jiang et al., 2021) uses intermediate layers for multi-depth localization.

Occlusion (Zeiler & Fergus, 2014) slides a mask over the input, measuring output changes to identify important regions.

RISE (Petsiuk et al., 2018b) applies random masks, evaluates output changes, and aggregates them into a weighted saliency map.

In our experiments, we certify all 12 previously discussed attribution methods which span all three categories.

**Robustness issues** A common problem of attribution methods is that they are not robust to small changes in the input. Many attack vectors exist, including adversarial inputs, model manipulations, backdoor attacks, and data poisoning (Baniecki & Biecek, 2024). In this work, we focus on small, imperceptible changes to the input that change the explanation while the model prediction remains unaffected. That is, for a model $f$, attribution method $h_f$, and input $x$, $f(x) = f(x')$ but $h_f(x) \neq h_f(x')$. It has a long line of work (Ghorbani et al., 2019; Kindermans et al., 2019; Dombrowski et al., 2019; Subramanya et al., 2019; Kuppa & Le-Khac, 2020; Zhang et al., 2020) and is the element that is most likely influenced by perturbations.

Several empirical defenses exist to make models and explanations more robust against input perturbations. A prominent line of work (Chen et al., 2019; Wang et al., 2020; Dombrowski et al., 2022; Tang et al., 2022) uses regularization techniques to improve attribution robustness of gradient-based attribution methods. For model agnostic approaches such as SHAP and LIME, there is work on making the sampling more robust (Ghalebikesabi et al., 2021; Shrotri et al., 2022; Vreš & Robnik-Šikonja, 2024). A different approach is to detect perturbed inputs using anomaly detection (Carmichael & Scheirer, 2023).

In our work, we address the vulnerability of attribution outputs to small, imperceptible noise by providing guarantees that $\ell_2$-bounded perturbations within a certified radius will not alter the certified attribution map.

**Certified explanations** While these defenses can empirically improve the robustness of certain explanations, they lack theoretical guarantees. Recognizing the importance of such guarantees, recent work has started to explore certifiably robust attributions. They typically use randomized smoothing to derive bounds on a per-image similarity score between perturbed and non-perturbed attributions.

The first work to consider such guarantees by Levine et al. (2019) defines a top-k score that measures the percentage of intersection between the k largest Sparsified SmoothGrad attributions of the base and the perturbed image. Sparsified SmoothGrad is defined as the expected value of sparsified Grad with added noise to the input. The authors compute a certified lower bound to this top-k similarity under bounded perturbations using Randomized Smoothing (Cohen et al., 2019; Lecuyer et al., 2019; Anani et al., 2024). Liu et al. (2022) propose a tighter bound using Rèni differential pri-

vacy. They also generalize the concept from $\ell_2$-norm perturbation models to general $\ell_p$-norm radii using generalized normal distributions. Expanding on top-k score, Chen et al. (2022) prove bounds on the ranking stability (thickness) of the top-k predictions.

Other methods consider bounds to the local Lipschitz constant (Tan & Tian, 2023) or $\ell_p$-norms (Wang & Kong, 2023; 2024) as robustness metrics. Lin et al. (2024) show an upper bound on the distance between perturbed and unperturbed removal-based attribution methods, and Dhurandhar et al. (2024) invert the typical certification process and grow robust attribution regions in the input space that maintain a certain level of fidelity.

We propose a novel approach to certify attribution robustness at the pixel level, producing high-quality visual certificates. Unlike prior work focused on non-interpretable image-level bounds, this is, to our knowledge, the first to offer pixel-level certification.

## 3. Preliminaries: Randomized Smoothing

This section introduces Randomized Smoothing for classification and segmentation, which is an essential background for our work. One key insight into our method is that we view attribution methods for classifiers as a segmentation task, which allows us to certify their pixel-wise robustness.

### 3.1. Classification

The core idea of Randomized Smoothing (Lecuyer et al., 2019; Cohen et al., 2019) is to transform any black-box classifier $f$ into a smoothed version $g$ by adding isotropic Gaussian noise to the input $x$:

$$g(x) := \arg\max_{c \in \mathcal{Y}} \mathbb{P}(f(x + \epsilon) = c), \qquad (1)$$

where $f : \mathbb{R}^m \to \mathcal{Y}$, $g : \mathbb{R}^m \to \mathcal{Y}$, and $\epsilon \sim \mathcal{N}(0, \sigma^2 I)$. The smoothed model $g$ therefore returns the class that $f$ predicts with the highest probability. Cohen et al. (2019) show that this prediction is robust (i.e., predicts the same class) for all inputs within an $\ell_2$-radius around the input $x$:

$$g(x) = g(x + \delta), \ \forall \delta \in \mathbb{R}^m, ||\delta||_2 \leq R, \qquad (2)$$

where the radius $R := \frac{\sigma}{2}(\Phi^{-1}(\underline{p_A}) - \Phi^{-1}(\overline{p_B}))$. $\Phi^{-1}$ denotes the inverse cumulative distribution function (CDF) of the standard Gaussian. $R$ depends on the noise level $\sigma$ and the difference between the lower bound of the top class $\underline{p_A}$ and upper bound of the runner-up class $\overline{p_B}$ probabilities.

Since $g(x)$ cannot be computed directly for general black-box functions $f$, Cohen et al. (2019) approximate it through Monte-Carlo sampling. Drawing $n$ i.i.d samples from $\mathcal{N}(0, \sigma^2 I)$, $g(x)$ is approximated with confidence $1 - \alpha$,

where $\alpha \in [0, 1)$ is a preset hyper-parameter that bounds the type I error probability.

## 3.2. Segmentation

Fischer et al. (2021) extend Randomized Smoothing to segmentation models $f : \mathbb{R}^{m \times N} \to \mathcal{Y}^N$ with $N$ output components (one prediction per pixel). The method strictly certifies stable components while abstaining from non-robust ones. To this end, the authors define a threshold for the top class probability $\tau \in [0.5, 1)$ and abstain for all components whose top class probability is below $\tau$. The smooth model $g^\tau : \mathbb{R}^{N \times m} \to \hat{\mathcal{Y}}^N$ is defined as:

$$g_i^\tau(x) = \begin{cases} c_{A,i} & \text{if } \mathbb{P}_{\epsilon \sim \mathcal{N}(0,\sigma^2 I)}(f_i(x+\epsilon) = c_{A,i}) > \tau, \\ \oslash & \text{otherwise,} \end{cases}$$
(3)

where $c_{A,i} = \arg\max_{c \in \mathcal{Y}} \mathbb{P}_{\epsilon \sim \mathcal{N}(0,\sigma^2 I)}(f_i(x+\epsilon) = c)$ and $\hat{\mathcal{Y}} = \mathcal{Y} \cup \{\oslash\}$ is the set of class labels combined with the abstain label. For all $g_i^\tau(x) \neq \oslash$, i.e., all pixels for which the method does not abstain, we get a guarantee that they are robust in an $\ell_2$ ball around the input $x$:

**Theorem 3.1** (from (Fischer et al., 2021)). *Let $\mathcal{I}_x = \{i \mid g_i^\tau(x) \neq \oslash, i \in 1, \ldots, N\}$ be the set of certified component indices in $x$. Then, for a perturbation $\delta \in \mathbb{R}^{m \times N}$ with $||\delta||_2 < R := \sigma \Phi^{-1}(\tau)$, for all $i \in \mathcal{I}_x$: $g_i^\tau(x+\delta) = g_i^\tau(x)$.*

## 4. Pixel-Certified Attributions

**Motivation** Attribution methods assign a real-valued score to each pixel, indicating its relevance to the classifier's prediction. Our goal is to certify the robustness of these pixel-level attributions – showing that for any attribution method, given an input image, pixels classified as "important" or "not important" robustly remain the same for an $\ell_2$ ball around the input. While prior work provides image-level bounds on similarity between perturbed and unperturbed attributions (Wang & Kong, 2023; 2024; Dhurandhar et al., 2024; Levine et al., 2019; Liu et al., 2022; Chen et al., 2022), no method certifies individual pixels in attribution maps. This gap likely stems from the challenge of certifying high-dimensional outputs. However, certifying a high-dimensional output was addressed by Fischer et al. (2021) via Randomized Smoothing for segmentation functions (Section 3.2). We thus aim to adapt attribution methods to such a framework to certify their outputs pixel-wise.

**Our approach** To achieve pixel-level certification of attribution methods, we reformulate the base attribution task as a segmentation problem, where each pixel is classified as either "important" or "not important". We achieve this by (i) sparsifying (i.e., binarizing) the output of the attribution function (Section 4.1) and (ii) constructing a smoothed ver-

sion of the sparsified attribution function (Section 4.2). This transformation allows us to view the smoothed sparsified attribution as a smoothed segmentation function (Section 4.3), enabling direct application of Theorem 3.1 for pixel-wise certification of the attribution output.

## 4.1. Sparsification

Sparsifying attribution outputs has been shown to improve their certifiable robustness (Liu et al., 2022). While sparsification does not reflect the absolute values of attributions, it suggests that relative ranks of attribution values are important for interpretation. We therefore operate on sparsified attribution functions, which (i) allows us to get theoretical guarantees on its outputs through Randomized Smoothing for segmentation, while (ii) producing high-quality certified attribution maps (Section 7.1).

Given an attribution function $h : \mathbb{R}^{c \times N} \to \mathbb{R}^N$, which maps an image $x \in \mathbb{R}^{c \times N}$ with $N$ pixels ($c = 3$ for RGB) to $N$ attribution real values, we define the sparsified version of this function $h^K : \mathbb{R}^{c \times N} \to \{1, 0\}^N$. It is parameterized by $K \in [50, 100]$, indicating that the top $K\%$ of values are mapped to 1 and the rest to 0:

$$h_i^K(x) = \begin{cases} 1 & \text{if } \frac{\text{rank}(h_i(x))}{N} \leq \frac{K}{100} \\ 0 & \text{otherwise} \end{cases}$$
(4)

where $\text{rank}(h_i(x))$ returns the descending order of the $i$-th pixel (e.g., 0 for largest, $N - 1$ for smallest).

## 4.2. Smoothed Sparsified Attributions

To certify the output of the sparsified attribution function $h^K$ in Eq. 4, we view it as a segmentation function that generates a segmentation map, assigning a class to each pixel. Specifically, $h^K$ classifies every pixel into two classes: 1 "important" or 0 "not important". We use Randomized Smoothing for segmentation as outlined in Section 3.2 to construct a smoothed version of a base sparsified attribution (segmentation) function. This function then provides per-pixel certified classes with robustness guarantees.

Specifically, given a sparsified attribution function $h^K$ and Gaussian noise $\epsilon \sim \mathcal{N}(0, \sigma^2 I)$, the smoothed sparsified attribution function $\bar{h}^{\tau,K} : \mathbb{R}^{c \times N} \to \{1, 0, \oslash\}$ parametrized by $\tau$ and the sparsification parameter $K$, is defined as:

$$\bar{h}_i^{\tau,K}(x) = \begin{cases} 1 & \text{if } \mathbb{P}_{\epsilon \sim \mathcal{N}(0,\sigma^2 I)}(h_i^K(x+\epsilon) = 1) > \tau \\ 0 & \text{if } \mathbb{P}_{\epsilon \sim \mathcal{N}(0,\sigma^2 I)}(h_i^K(x+\epsilon) = 0) > \tau \\ \oslash & \text{otherwise.} \end{cases}$$
(5)

which returns the certified class 1 or 0 for the $i$-th pixel if the class probability is higher than the threshold $\tau \in [0.5, 1)$, otherwise, it abstains. A pixel certified as "1" means that

it remains in the top $K\%$ of values across all perturbed attributions with a high probability, while "0" means that it remains in the bottom $(100\text{-}K)\%$. For all non-abstain indices in the smoothed sparsified attribution function $\bar{h}^{\tau,K}$ output, $\bar{h}_i^{\tau,K}(x) \neq \oslash$, the certified class remains the same within an $\ell_2$ ball around the input $x$, as stated in Theorem 3.1.

The certified radius $R := \sigma\Phi^{-1}(\tau)$ from Theorem 3.1 of the smoothed sparsified attribution function is high when (i) the noise level $\sigma$ is high, but it comes at the cost of a degraded performance in the base model, and (ii) the top class probability $\tau$ is high, as it is a more strict condition to certify the top class. We analyze the impact of varying the certified radius in Sections 7.2, 7.3 and App. B, qualitatively in App. E and G, and the effect of $\tau$ in App. C.

### 4.3. Estimating the smoothed sparsified attribution

Evaluating the output of the sparsified smoothed attribution $\bar{h}^{\tau,K}$ in Eq. 5 requires sampling from the sparsified attribution $h^K$ Eq. 4 as it is evaluated on a data distribution $X := \mathcal{N}(x, \sigma^2 I)$ around $x$ instead of a single point $x$. By design, the smoothed sparsified attribution $\bar{h}^{\tau,K}$ is mathematically equivalent to the smoothed segmentation model $g^\tau$ in Eq. 3. We thus employ the same Monte-Carlo sampling used for smoothed segmentation to compute the certified attribution output (Fischer et al., 2021).

## 5. Robustness Evaluation Metrics

To evaluate the certified robustness, localization, and faithfulness of attribution methods, we introduce three quantitative metrics.

**%certified** In our pixel-level certification setup, a pixel in the output is either certified or abstained from. Abstentions occur when the assigned class for the pixel fluctuates in noise during sampling, indicating that the pixel does not consistently belong to the top $K\%$ or bottom $(100\text{-}K)\%$ of perturbed attributions. Therefore, we use %certified as a robustness metric, which is the ratio of certified pixels to all pixels. Given an image $x \in \mathbb{R}^{c \times N}$, %certified is defined as:

$$\%\text{certified}(x) = \frac{|\{i \mid \bar{h}_i^{\tau,K}(x) \neq \oslash\}|}{N} \quad (6)$$

This score is maximized when all pixels are certified $(\%\text{certified}(x) = 1)$ and minimized when all pixels are abstained from $(\%\text{certified}(x) = 0)$.

**Certified GridPG** Since there is no ground truth to the attribution methods, several metrics have been proposed to evaluate them. Localization metrics measure how well the attribution method localizes class-discriminative features. For the Grid Pointing Game (GridPG) (Bohle et al., 2021), the attribution methods are evaluated on a synthetic $m \times m$

grid of images in which every class occurs at most once. GridPG then measures the ratio of the positive attributions assigned to a subimage within the grid to the rest of the grid. Let $A^+(p)$ refer to the positive attribution for the $p$-th pixel in the grid. Then, the localization score for a subimage $x_i$:

$$L_i = \frac{\sum_{p \in x_i} A^+(p)}{\sum_{j=1}^{m^2} \sum_{p \in x_j} A^+(p)} \quad (7)$$

This score is maximized when all positive attributions are fully localized within a subimage, yielding a score of $L_i = 1$, and minimized when no positive attributions are located within the subimage $L_i = 0$.

Instead of using positive attributions, we reformulate this metric to find the ratio of pixels certified as "1" (i.e., to be in the top $K\%$ of the grid) within the subimage to the whole grid. Let $A^c(p)$ be a binary function that only returns 1 for pixels certified as top $K\%$. Then, the certified localization score for a subimage $x_i$ is:

$$cL_i = \frac{\sum_{p \in x_i} A^c(p)}{\sum_{j=1}^{m^2} \sum_{p \in x_j} A^c(p)} \quad (8)$$

Both metrics are certified, which means their values remain the same within an $\ell_2$ ball around the input bounded by the certified radius.

**Faithfulness** Beyond robustness and localization, we also assess whether certified attributions truly highlight class-relevant evidence. Inspired by deletion metrics used in prior work on explanation evaluation (Fong & Vedaldi, 2017; Petsiuk et al., 2018a), we define a deletion-based faithfulness score for certified attributions. For a given image and its certified attribution maps at sparsification levels $K$, we iteratively remove (e.g., set to zero) the pixels that are certified as "1" – starting with the most important (lowest $K$) – and record the model's confidence in the original class after each removal step. A steep confidence drop after deleting a small fraction of certified pixels indicates that those pixels were indeed crucial to the prediction, signaling high faithfulness.

## 6. Experimental Setup

**Dataset and Architecture** We use the ImageNet (Russakovsky et al., 2015) dataset and 5 classification models: ResNet-18 (He et al., 2016), Wide ResNet-50-2 (Zagoruyko & Komodakis, 2016), ResNet-152 (He et al., 2016), VGG-11 (Karen, 2015), and ViT-B/16 (Dosovitskiy et al., 2020). Implementation details for extending attributions from CNNs to transformers (e.g., ViT-B/16) are in App. A.2. We select images that were classified with a high confidence by ResNet-18 and VGG-11, ensuring the top $K\%$ pixels constitute positive evidence for the correct class. Due to the

computational cost of sampling during certification, we randomly select 100 images from the validation set, on which we compute all results. The image dimension is $224 \times 224$.

**Evaluation on Certified GridPG** We create $2 \times 2$ grids of images from distinct classes, sampled from the confidence-filtered validation set. For each attribution method, we generate 100 such grids to compute GridPG and Certified GridPG scores. Each grid has a resolution of $448 \times 448$.

**Evaluation at layers** We evaluate all attribution methods at the input and final spatial layer for a fair comparison (Rao et al., 2022). Attributions from deeper layers are bilinearly upsampled to input resolution (Selvaraju et al., 2017).

**Attribution methods** We certify 12 attribution methods from 3 families: (i) **backpropagation-based:** Grad (Simonyan, 2014), GB (Springenberg et al., 2014), IntGrad (Sundararajan et al., 2017), IxG (Shrikumar et al., 2017) and LRP (Bach et al., 2015) (ii) **activation-based:** CAM (Zhou et al., 2016), Grad-CAM (Selvaraju et al., 2017), Grad-CAM++ (Chattopadhay et al., 2018), Ablation-CAM (Ramaswamy et al., 2020) and Layer-CAM (Jiang et al., 2021), and (iii) **perturbation-based:** RISE (Petsiuk et al., 2018b) and Occlusion (Zeiler & Fergus, 2014). The implementation details are in App. A.1. We display SS (Smoothed Sparsified), which is the average of the sparsified attributions evaluated on $n = 100$ noisy input samples per image. Note that LRP and Cam are not evaluated on ViT-B/16 due to the extension complexity to the transformer architecture.

**Certification** We use noise level $\sigma = 0.15$, number of samples $n = 100$ per image, top class probability $\tau = 0.75$ and $\alpha = 0.001$, unless stated otherwise. All certified results are robust with confidence $1 - \alpha$ w.r.t the radius $R = 0.10$, unless stated otherwise. Abstain pixels are marked gray (labelled ⊘), and certified bottom $(100 - K)\%$ ("0") pixels are marked white, and top $K\%$ ("1") are colored otherwise.

# 7. Results and Discussion

We evaluate our certification approach across three attribution families, presenting qualitative certified attribution visuals (7.1) and quantitative robustness comparisons using %certified (7.2) and Certified GridPG (7.3). We also analyze the robustness-localization tradeoff (7.4), highlighting methods that have most balance, and their faithfulness (7.5).

## 7.1. Certified Attributions Visuals

Figure 2 shows certified attribution maps at different sparsification values $K$. As $K$ decreases, fewer pixels are certified as top $K\%$ ("1"), highlighting finer details (e.g., LRP marks the fish's eye at $K = 5\%$), while larger $K$ values capture coarser features (e.g., LRP highlights the full fish

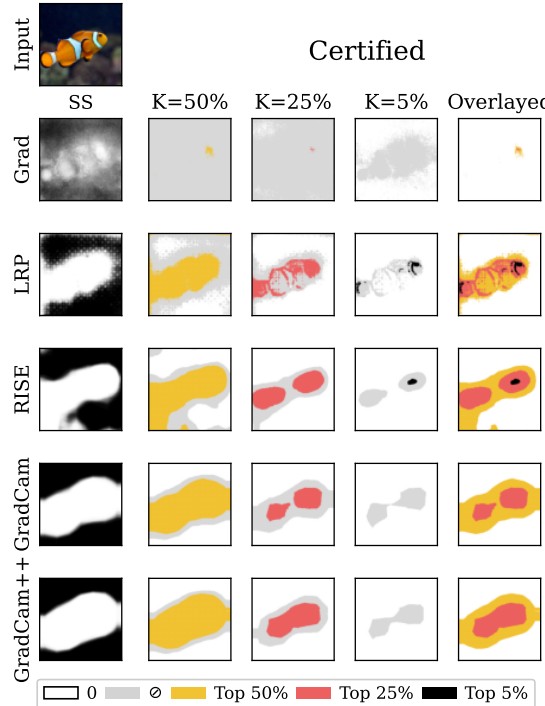

Figure 2: **Certified attributions of selected methods at different sparsification parameter $(K)$ values on ResNet-18.** SS (Smoothed Sparsified) is evaluated on $n = 100$ noisy input samples per image and at $K = 50\%$. The "Overlayed" column shows overlayed certified top $K\%$ pixels per method, with lower $K$ taking precedence. Extended figures for ResNet-18 and ResNet-152 on all attribution methods at various certified radii for more examples are in App. E.

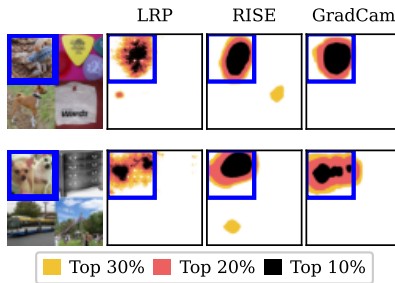

Figure 3: **Examples from overlayed certified GridPG attributions at different sparsification $K$ values on ResNet-18.** The blue square denotes the ground truth subimage.

at $K = 50\%$). Overlaying maps across $K$ (see "Overlayed" column) reveals a pixel importance hierarchy, with darker pixels denoting higher importance. Figure 4 extends this to more examples and all 12 methods. Certified attributions highlight semantic features well, with LRP offering the most detail. Input-layer methods (backpropagation and perturbation) benefit from higher $K$ due to noise-induced ranking instability, while activation methods benefit from a lower $K$ for finer isolation of their coarse outputs.

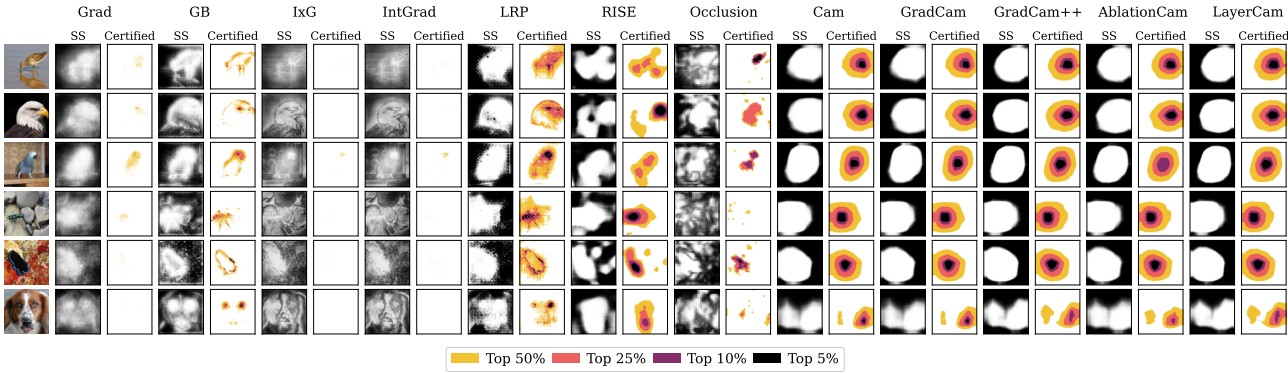

Figure 4: **Overlayed certified attributions at different $K$ values across methods on ResNet-18.** SS (Smoothed Sparsified), which refers to the average of the sparsified attributions, is evaluated on $n = 100$ noisy input samples per image and at $K = 50\%$. This figure is extended to ViT-B/16 in App. Figure 20.

To visualize the localization of certified attributions, Figure 3 displays certified attribution grid examples. In each grid, the top-left subimage contains the certified top $K\%$ pixels, which are correctly localized to the target class. A rigorous qualitative analysis of certified grids is in App. G.

### 7.2. Robustness Evaluation of Attributions

We assess attribution robustness using the %certified metric in Figure 5 at the input and final layers across certified radius $R$ and sparsification $K$ values. Increasing $R$ (Figure 5, top) reduces certification rates for most methods, as it increases the noise $\sigma$, making attributions less stable. Grad and GB at the final layer are exceptions, as they produce almost constant grid-like patterns in ResNet-18, though uninformative (Section 7.3). Lowering $K$ (Figure 5, bottom) shifts certification rates from top $K\%$ pixels (darkest shade)

to lower $(100 - K)\%$ pixels (medium shade). At the final layer, methods certify more top-$K\%$ pixels, reflecting the robustness of high-level features to noise. We next discuss results by method family.

**Backpropagation methods** At the input layer (Figure 5, top), LRP is the most robust among backpropagation methods (second overall after RISE), while other methods show %certified $\approx 0$, due to their reliance on input magnitude (e.g., IxG and IntGrad), hence increasing sensitivity to perturbations.

**Activation-based methods** In Figure 5 (top), activation methods outperform backpropagation methods in %certified at the final layer as $R$ increases, with the exception of Grad and GB. Grad-CAM++ and Layer-CAM score higher %certified than other activation methods.

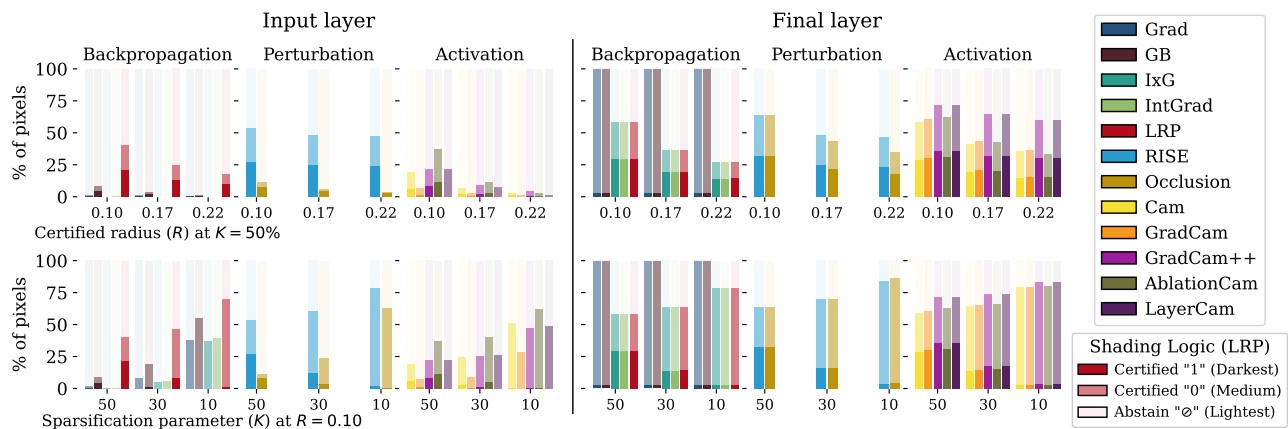

Figure 5: **Comparison of the per-pixel certification rate (%certified) on ResNet-18 across backpropagation, perturbation and activation methods.** (*Left*) shows evaluation at the input and (*Right*) at the final layer using different certified radii $R$ (*Top*) and sparsification parameter $K$ values (*Bottom*). The darkest shades denote %certified pixels, while brightest denotes abstain $\oslash$. This figure is extended to 5 models in App. Figures 9 and 10.

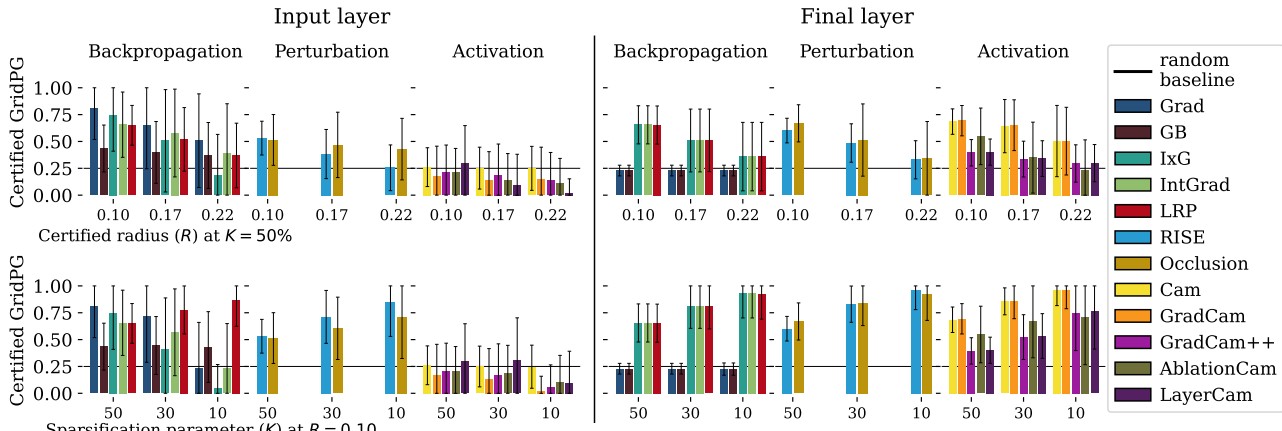

Figure 6: **Comparison of the certified localization (Certified GridPG) on ResNet-18 across backpropagation, perturbation and activation methods**. (*Left*) shows evaluation at the input and (*Right*) at the final layer using different certified radii (*Top*) and sparsification parameter $K$ values (*Bottom*). This figure is extended to 5 models in App. Figures 11 and 12

**Perturbation-based methods** RISE outperforms all other methods at the input layer, with the highest certification rates across all radii. It has a high %certified score at both layers, unlike Occlusion, which improves only at the final layer. This highlights the robustness advantage of randomized over deterministic masking.

> **Certified Robustness**: RISE and LRP are the most robust across settings. All methods are more robust on the final layer.

### 7.3. Certified Localization Evaluation of Attributions

We evaluate the localization performance of attribution methods using the Certified GridPG metric (Eq. 8) at the input and final layers in Figure 6 by varying the certified radius $R$ and sparsification parameter $K$. Increasing $R$ raises the variance in Certified GridPG scores, with RISE showing the least variance. Activation methods perform worse than random on the input layer. LRP, RISE and Occlusion acheive near perfect localization at $K = 10\%$ (Figure 6, bottom), suggesting that certifying fewer top $K\%$ pixels enhances focus on the correct subimage in the grid in methods that are already robust (see Figure 5). Overall, perturbation methods (e.g., RISE) demonstrate superior localization as $R$ increases or $K$ decreases at both layers, making them more reliable. We next discuss results by method family.

**Backpropagation methods** At $R=0.10$ (Figure 6, left), Grad achieves the highest Certified GridPG at the input layer, followed by IxG, IntGrad and LRP; GB performs worst. At the final layer, Grad and GB drop to random localization despite high certification rates (Figure 5), showing that robustness does not guarantee localization (Section 7.4).

**Activation methods** At the final layer, increasing $R$ lowers

Certified GridPG below the random baseline for all but CAM and Grad-CAM, scoring the highest. Reducing $K$ improves localization across methods, with CAM and Grad-CAM achieving near-perfect performance at $K = 10\%$.

**Perturbation methods** RISE and Occlusion have comparable Certified GridPG scores at both the input and final layer. As $K$ decreases at the input layer, RISE and Occlusion improve, with RISE achieving a near-perfect score at $K = 10\%$.

> **Certified Localization**: LRP, RISE, and Occlusion localize best across settings. Except for Grad and GB, all methods localize well at the final layer.

### 7.4. Robustness vs. Localization Tradeoff

We previously analyzed certified robustness (Section 7.2) and localization (Section 7.3), noting that higher robustness does not necessarily imply better localization and vice versa. To put both together, Figure 7 examines the tradeoff between certified robustness (%certified) and localization (Certified GridPG) across 5 models including CNNs and a ViT.

**Input layer** RISE and LRP offer the best tradeoff across models, with high localization and robustness. Grad, GB, and IntGrad score highest on Certified GridPG (e.g., W-ResNet-50-2, ResNet-152) but show near-zero robustness. Activation methods perform poorly on both metrics.

**Final layer** RISE again strikes the best balance alongside Occlusion and activation methods, while Grad and GB are most robust but show random-like localization. RISE maintains the best balance especially on ViT-B/16 where others fail. Overall, the tradeoff improves at the final layer, suggesting coarser attributions better enhance localization and robustness, aligning with results in 7.2 and 7.3.

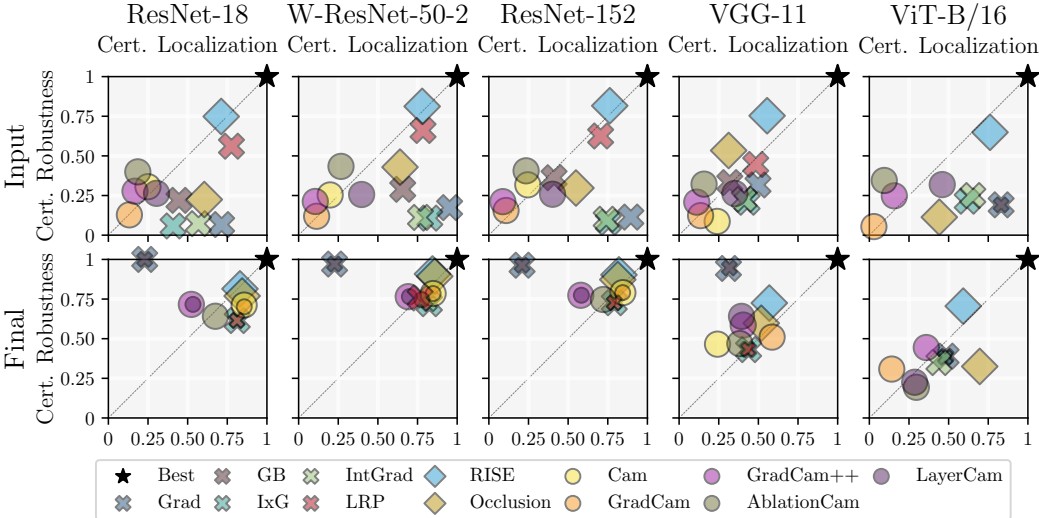

Figure 7: **Robustness-localization tradeoff of attribution methods using the %certified and Certified GridPG metrics on all 5 models at** $K = 30\%$. (*Top*) evaluation is on input and (*Bottom*) final layers.

## 7.5. Faithfulness Analysis of Certified Attributions

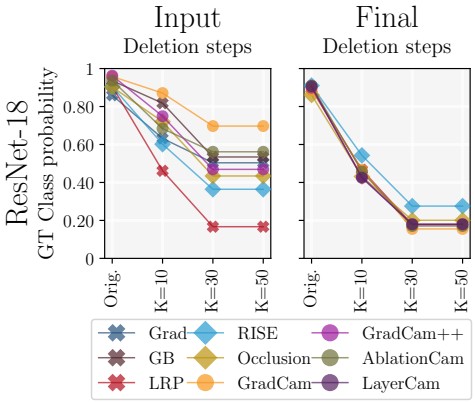

Figure 8: **Faithfulness comparison of certified attribution methods on ResNet-18** using the ground truth (GT) class confidence against the deletion steps in which top $K$ certified pixels are removed in descending order of importance. (*Left*) evaluation is at the input and (*Right*) at the final layer.

We assess the faithfulness of certified attributions using a deletion-based (Petsiuk et al., 2018b) analysis (described in Section 5) in Figure 8.

**Input layer** In Figure 8 (left), LRP and RISE cause the steepest drop in class confidence after the first deletion step, indicating high faithfulness. Grad-CAM yields the smallest decrease, reflecting weak alignment with class-relevant features, consistent with its near-random certified localization in Figure 6. LRP and RISE also achieve the highest Certified GridPG scores across models (Figures 6, 7), showing strong alignment between certified localization and faithfulness.

**Final layer** In Figure 8 (right), most methods induce a prediction flip after the first step, indicating higher faithfulness than at the input layer. However, some show high faithfulness despite poor localization. For example, GB and Grad yield near-random Certified GridPG scores (Figure 6) but cause steep confidence drops (Figure 8). This suggests their grid-like final-layer patterns introduce deletion artifacts that disrupt predictions without identifying relevant regions.

All three metrics, %certified, Certified GridPG and faithfulness, complement each other to understand different aspects of certified attributions, which is essential for producing trustworthy attributions in safety-critical domains.

> **LRP** and **RISE** strike the best balance among **robustness**, **localization** and **faithfulness**, across settings.

## 8. Conclusion

We propose pixel-wise certification of attribution methods and a framework to evaluate their certified robustness and localization. First, we introduce per-pixel certification for smoothed sparsified attributions by reformulating attribution as a segmentation task, enabling direct application of Randomized Smoothing to produce certified, high-quality attribution maps. Second, we present an evaluation scheme using three metrics, percentage of certified pixels, Certified GridPG, and deletion-based faithfulness, to evaluate twelve attribution methods across backpropagation, activation, and perturbation families. Our quantitative analysis shows that RISE and LRP achieve the best tradeoff across the three metrics across settings, while other methods vary in performance.

## Impact Statement

We introduce a novel framework for certifying attribution methods at the pixel level, reframing attribution as a segmentation task and applying Randomized Smoothing for segmentation to certify each pixel as "important" or "not important" under $\ell_2$-bounded input perturbations. This yields the first certifiably robust attribution maps applicable to any black-box method, offering both fine-grained visual explanations and per-pixel robustness guarantees.

Our approach enables two key contributions: (i) robust, interpretable certified attributions that can inform downstream tasks, and (ii) a new evaluation paradigm for attribution robustness, including visual and quantitative comparisons across methods. To this end, we propose %certified, which measures robustness, and Certified GridPG, which captures robust localization.

This work addresses the critical lack of robustness in attribution methods and underscores the ethical and societal importance of trustworthy AI explanations in high-stakes settings such as healthcare, justice, and autonomous systems.

**Societal Impact** Our framework advances trustworthy and explainable AI by providing certified attributions that are reliable under input perturbations. In domains like healthcare, law, and autonomous driving, where understanding model reasoning is vital, our method enhances decision transparency and reliability. It also introduces a principled way to compare attribution methods, fostering more robust and interpretable deep learning models. Collaborations with practitioners can support integration into real-world systems, increasing accountability and trust.

**Ethical Considerations** In safety-critical domains, especially medical decision-making, certifiably robust attributions offer ethical advantages by reducing the risk of misleading explanations. When paired with expert oversight, our method supports more informed and reliable decision-making.

**Conclusion** We present the first general-purpose framework for certifying and evaluating attribution methods at the pixel level, addressing long-standing robustness challenges. Our method enables practical, reliable, and trustworthy model explanations, and lays the foundation for future work in certifiably robust AI. Ultimately, we anticipate a positive societal impact by improving explainability, robustness, and safety in AI systems.

## Acknowledgements

We greatly thank Dr. Jonas Fischer for the valuable feedback and technical discussions during the rebuttal phase. We very much appreciate the diligent and constructive reviews by all reviewers, and believe the additional insights gained in preparing the rebuttal, and expanding the analysis in our work, significantly strengthen our paper.

This work was partially funded by ELSA – European Lighthouse on Secure and Safe AI funded by the European Union under grant agreement No. 101070617. Views and opinions expressed are however those of the authors only and do not necessarily reflect those of the European Union or European Commission. Neither the European Union nor the European Commission can be held responsible for them.

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

# A. Implementation Details

## A.1. Attribution Methods

**LRP (Bach et al., 2015)**   Following (Montavon et al., 2017; Arias-Duart et al., 2021; Rao et al., 2022), we use the configuration that applies the $\epsilon$-rule to the fully connected layers in the network, with $\epsilon = 0.25$, and the $z^+$-rule to the convolutional layers, except for the first convolutional layer where we use the $z^B$-rule.

**Occlusion (Zeiler & Fergus, 2014)**   Occlusion uses a sliding window of size $K$ and stride $s$ over the input. Following (Rao et al., 2022), we use $K = 16$, $s = 8$ for the input layer and $K = 5$, $s = 2$ for the final layer.

**RISE (Petsiuk et al., 2018b)**   We use a total of $N = 6000$ randomly generated masks per layer. Masks are initially generated on a low-resolution grid of size $s \times s$ (with default $s = 6$), where each cell is activated with probability 0.1. Each binary grid is bilinearly upsampled and cropped to match the input dimensions. The implementation supports CNN-like 4D tensors, following the setup in (Rao et al., 2022), as well as ViT-like 3D tensors.

## A.2. Vision Transformer ViT-B/16

While evaluating attribution methods at the input layer for Vision Transformers is standard, evaluating them at the final layer requires modifications to the attribution logic. In CNNs, the final layer contains activations in the form of a 3D tensor of shape $(C, H, W)$, where $C$ is the number of channels and $(H, W)$ correspond to spatial locations at a lower resolution compared to the input image. This structure aligns naturally with the input image and is usually interpolated (i.e., resized) to match its dimensions.

However, Vision Transformers (e.g., ViT-B/16 (Dosovitskiy et al., 2020)) process the image as a sequence of patches, with each patch treated as a token. The final feature representation prior to classification is typically of shape $(N, D)$, where $N$ is the number of tokens (including the [CLS] token), and $D$ is the hidden dimension (e.g., 768 for ViT-B/16). For ViT-B/16 and a $224 \times 224$ input image, this results in $N = 197$ tokens: 1 [CLS] token and 196 patch tokens, where each patch corresponds to a $14 \times 14$ grid in the original image (a result of dividing 224 by 16).

To enable attribution on ViTs, we discard the [CLS] token and reshape the remaining 196 tokens into a $(14, 14)$ grid, analogous to CNN feature maps. This allows us to treat patch embeddings as spatial features and apply attribution methods at the final layer similarly to how they are applied in CNNs. We adapt our attribution logic to extract these reshaped features, which are then bilinearly interpolated to the original image size.

We have extended all attribution methods to be ViT-compatible, with the exception of LRP (Bach et al., 2015) and CAM (Zhou et al., 2016), due to the lack of direct extensions for these methods.

# B. Evaluation of certified attributions on 5 models: ResNet-18, W-ResNet-50-2, ResNet-152, VGG-11 and ViT-B/16

## B.1. Certified Robustness (%certified)

We assess attribution robustness on five different models using the %certified metric evaluated at the input and final layers across certified radii ($R = 0.10, 0.17, 0.22$) in Figure 9 and sparsification ($K = 50, 30$ and 10) in Figure 10. Note that ViT-B/16 is not evaluated on CAM and LRP. The general trend of reduced certification rates by increasing the certified radius $R$, as well as reduced certified top $K\%$ pixels by decreasing $K$ holds across all five models.

**Input layer**   LRP and RISE exhibit the highest robustness, as they also maintain a relatively high %certified scores across all three radii values in Figure 9. Interestingly, they also maintain a balance in certified top $K\%$ pixels by decreasing $K$ in Figure 10.

**Final layer**   Though Grad and GB still seem the most robust on CNNs in Figure 9, they localize very poorly, this is because they produce grid-like almost constant attributions at the final layer of the CNN architecture, due to max-pooling.

**ViT-B/16**   Gradient-based methods exhibit low robustness on ViT-B/16 comparable to their CNNs performance on both the input and final layers in Figure 9. Interestingly, RISE maintains the same performance of having high robustness across

both architecture types, with the highest at radius $R = 0.10$ at the final layer on ViT-B/16. Activation-based methods perform poorly on the transformer model compared to CNNs on the final layer. This is likely due to the lack of spatially structured convolutional features in ViTs, which affects the quality and stability of activation-based attributions when applied to token-based representations. We discuss this implementation detail in App. A.2.

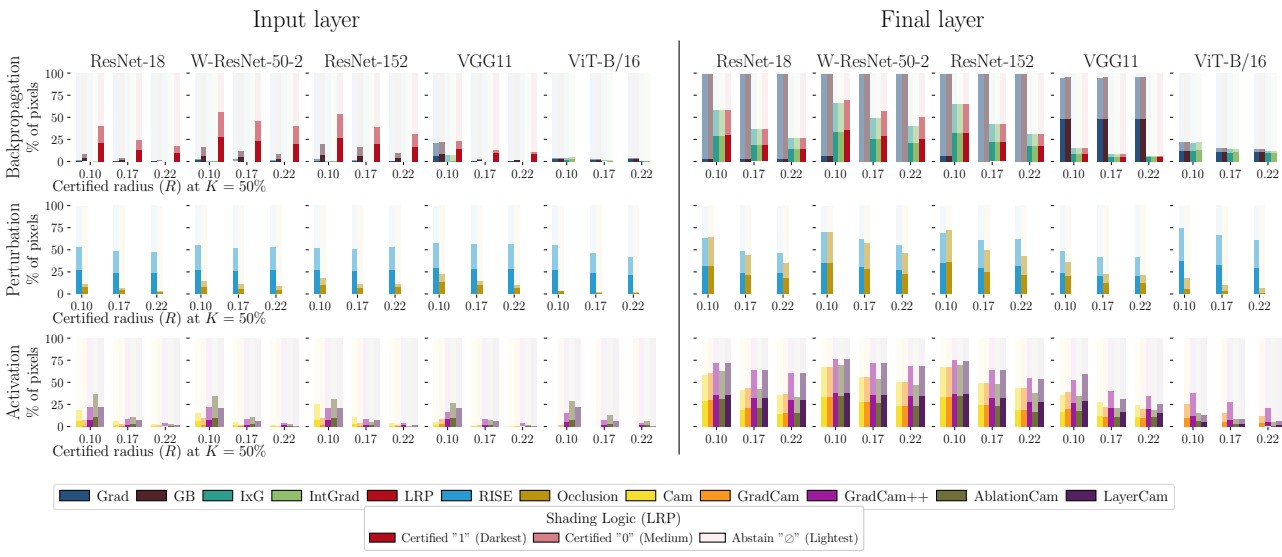

Figure 9: **Comparison of the per-pixel certification rate (%certified) against the certified radius $R$ on all 5 models across backpropagation, activation and perturbation methods.** (*Left*) shows evaluation at the input and (*Right*) at the final layer. The darkest shades denote %certified pixels, while brightest denotes abstain ⊘.

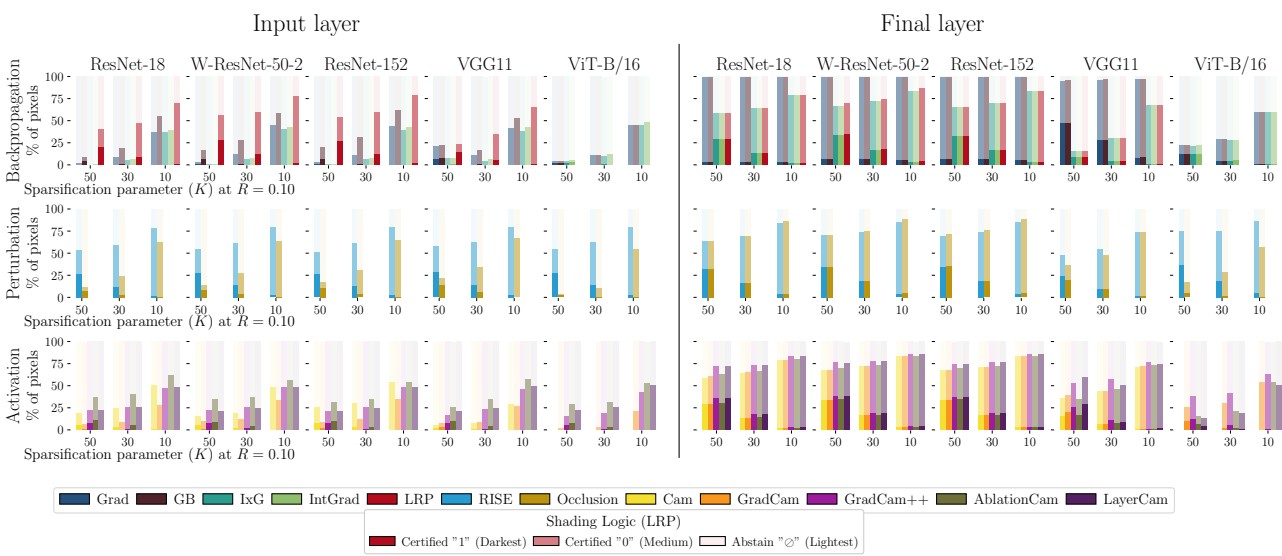

Figure 10: **Comparison of the per-pixel certification rate (%certified) against sparsification values $K$ on all 5 models across backpropagation, activation and perturbation methods.** (*Left*) shows evaluation at the input and (*Right*) at the final layer.

## B.2. Certified Localization (Certified GridPG)

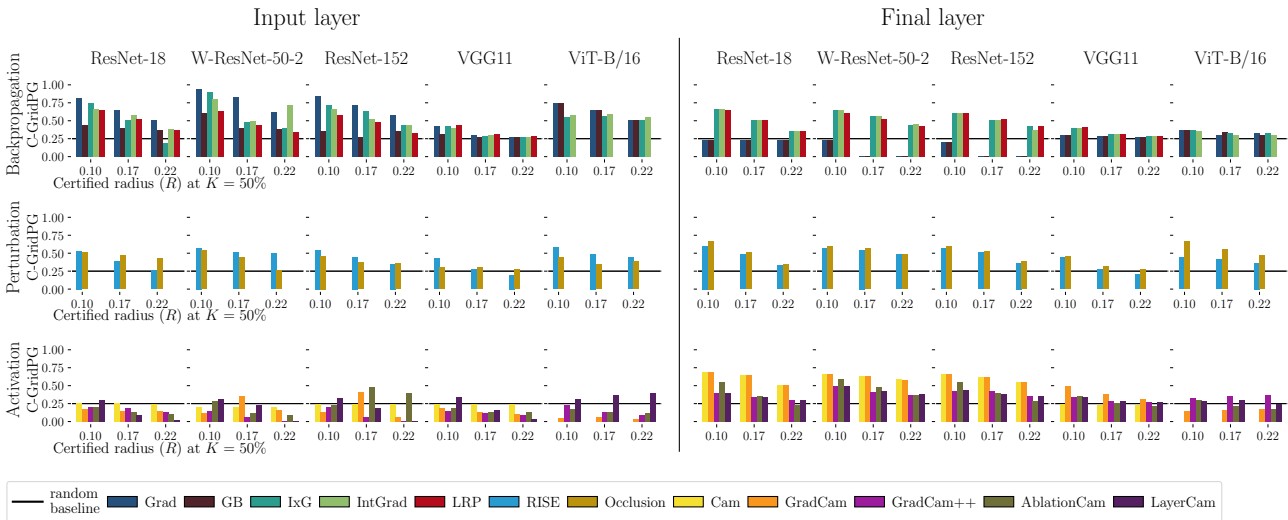

Figure 11: **Comparison of the certified localization (Certified GridPG) against the certified radius $R$ on all 5 models across backpropagation, activation and perturbation methods.** (*Left*) shows evaluation at the input and (*Right*) at the final layer.

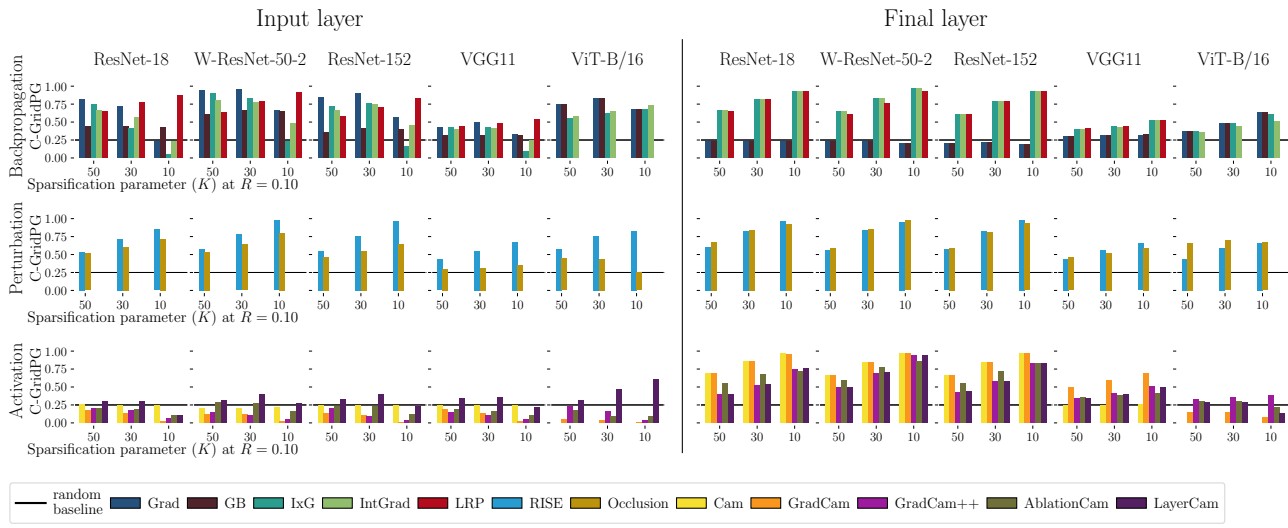

Figure 12: **Comparison of the certified localization (Certified GridPG) against the sparsification values $K$ on all 5 models across methods.** (*Left*) shows evaluation at the input and (*Right*) at the final layer.

**Input layer** In Figures 11 and 12, backpropagation and perturbation methods demonstrate effective localization at the input layer across all models, except VGG-11. This is likely due to the absence of skip connections in VGG-11, which hampers gradient flow and impedes signal propagation to the input. In contrast, activation-based methods perform worse than random across all models, as they rely solely on high-level forward activations from the final layers, lacking direct correspondence with input pixels.

**Final layer** At the final layer, all attribution methods generally yield higher localization scores compared to the input layer, as shown in Figures 11 and 12. This improvement is particularly evident in activation and perturbation methods, which align better with the spatial structure of final-layer features. However, exceptions are observed in Grad and Guided Backpropagation (GB), which underperform relative to other methods.

# C. Impact of varying the certification strictness (hyperparameter $\tau$)

One of the hyperparameters in our certification setup is $\tau \in (0.5, 1]$, which sets a threshold for the top class probability of a pixel to certify it (discussed in Section 4.2 and defined in Eq. 5). The higher the value of $\tau$, the more strict the certification condition is. We investigate the effect of increasing $\tau$ on %certified and Certified GridPG of all attribution methods in Figure 13.

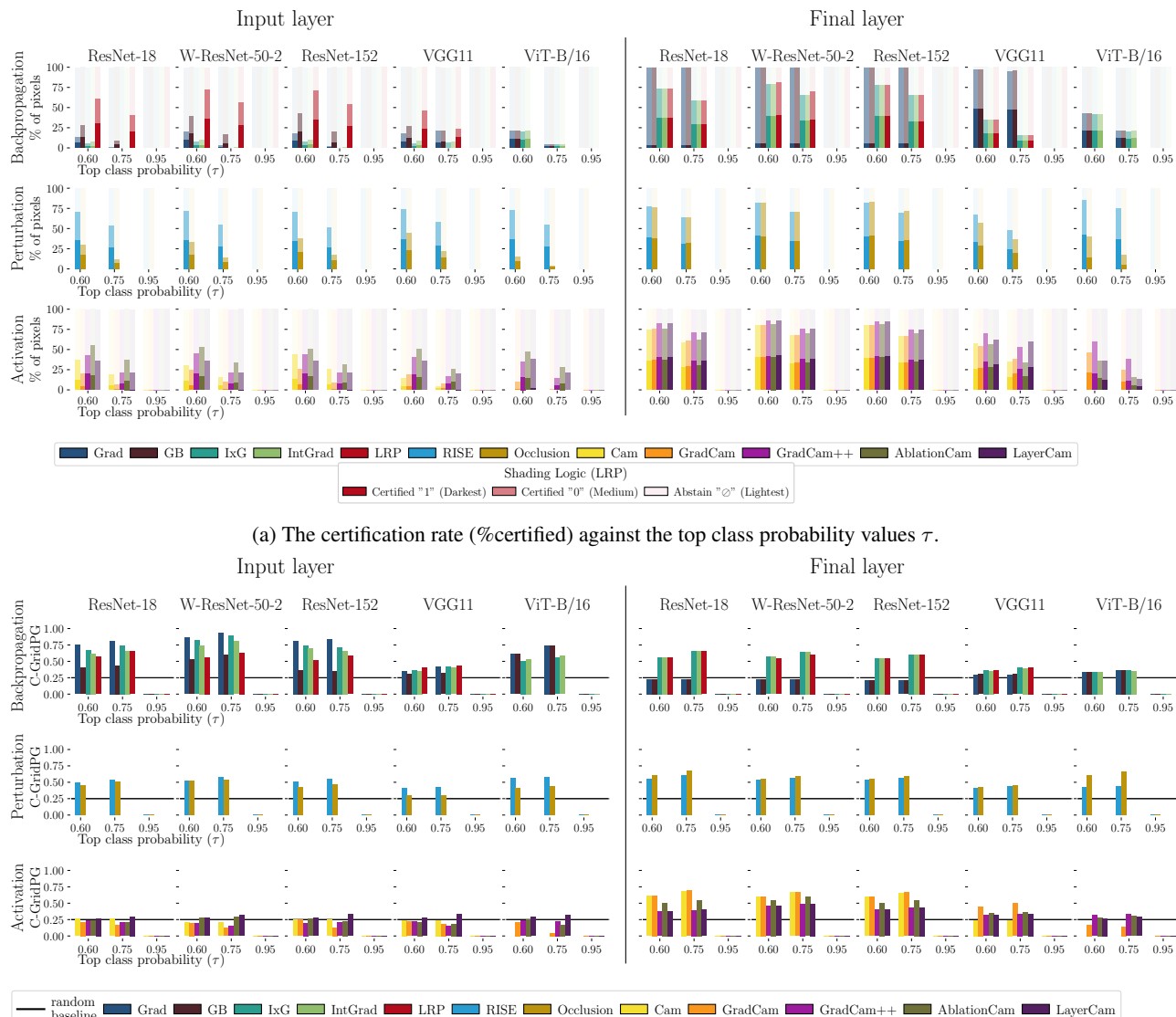

(a) The certification rate (%certified) against the top class probability values $\tau$.

(b) Certified GridPG against the top class probablity values $\tau$.

Figure 13: The performance of attribution methods in terms of %certified (a) and Certified GridPG (b) by increasing the top class probability $\tau$ (making the certification more strict)

**Effect of $\tau$ on %certified** In Figure 13, we observe that increasing $\tau$ lowers the certification rate of all methods, except for Grad and GB at the final layer, since they produce almost constant grid-like patterns at that layer in ResNet18. At the highest value of $\tau = 0.95$, the performance of all methods drops to 0, indicating for the need of increasing the number of samples to be able to certify under the strict condition imposed by a high $\tau$.

**Effect of $\tau$ on Certified GridPG**   In Figure 13 (b), interestingly, increasing $\tau$ from 0.60 to 0.75 boosts the localization performance of all methods (with the exception of activation methods completely failing at the input layer). This indicates that by imposing a more strict certification setup, only the most confident pixels are certified to top K% in attribution output, which also localizes better. Hence, we use a default value of $\tau = 0.75$ throughout the paper.

## D. GridPG and Certified GridPG

To understand how attribution methods maintain localization under input noise, we analyze the relationship between the original localization score (GridPG) and the certified localization score (Certified GridPG) in Figure 14. This comparison helps assess the robustness of each method's localization ability when subjected to noise during certification. At the input layer (Figure 14, top), LRP outperforms all methods, striking a balance in both metrics. Backpropagation-based and activation-based methods exhibit a higher Certified GridPG than their respective GridPG scores. This increase may be attributed to the certification process rejecting incorrect positive evidence located outside the correct subimage in the grid, thereby boosting the certified localization score. At the final layer (Figure 14, bottom), RISE is the only method that achieves a higher Certified GridPG than GridPG, demonstrating its ability to improve localization when subjected to input noise. Meanwhile, all other methods roughly lie on the diagonal, showing equal values of Certified GridPG and GridPG. GradCam achieves the highest scores in both GridPG and Certified GridPG (with the exception of ViT-B/16), indicating that its localization performance remains consistent between the raw and certified outputs.

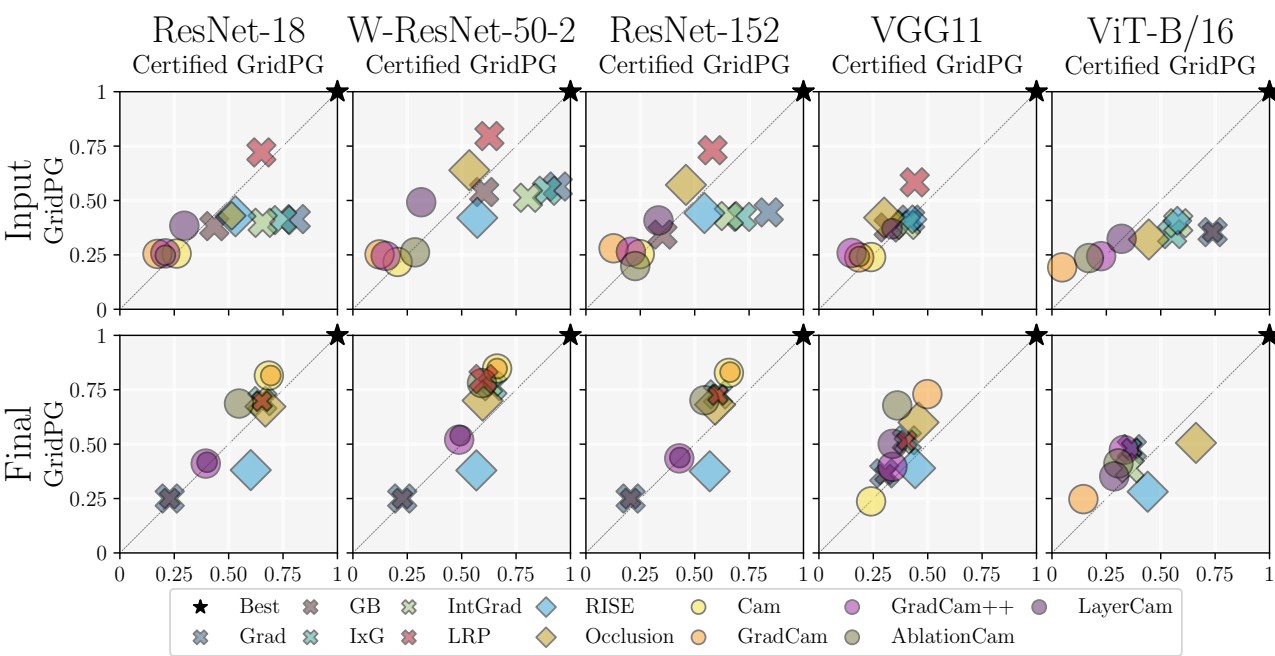

Figure 14: The performance of attribution methods on ResNet18 in terms of original GridPG score against Certified GridPG.

## E. Impact of varying sparsification and certified radius on certified visuals

We present qualitative examples showing the certified visuals on ResNet-18 (Figures 15 and 16) and additionally ResNet-152 (Figures 17 and 18) of all attribution methods by varying the sparsification parameter ($K = 50\%, 25\%, 10\%$ and $5\%$) and certified radius ($R = 0.10, 0.17, 0.22$).

As the radius $R$ increases with higher noise levels ($\sigma$), all methods abstain more and certify fewer top $K\%$ pixels. Amongst backpropagation-based methods, only LRP, Grad and GB show certified top K% pixels in the overlayed output, whilst IxG and IntGrad almost abstain from all top K% pixels at all radii. A notable observation is that activation-based methods show high-quality overlayed certified maps across all noise levels (certified radii) on both models ResNet-18 and ResNet-152. Perturbation-based methods maintian high-quality overlayed maps across all certified radii.

As $K$ decreases, fewer pixels are certified within the top $K\%$, while more pixels fall within the lower $(100 - K)\%$ across methods. At smaller $K$ values (e.g., $K = 5\%$), most methods highlight more distinctive features. Overlaying certified attribution maps across different $K$ values offers insights into the relative pixel importance for each method.

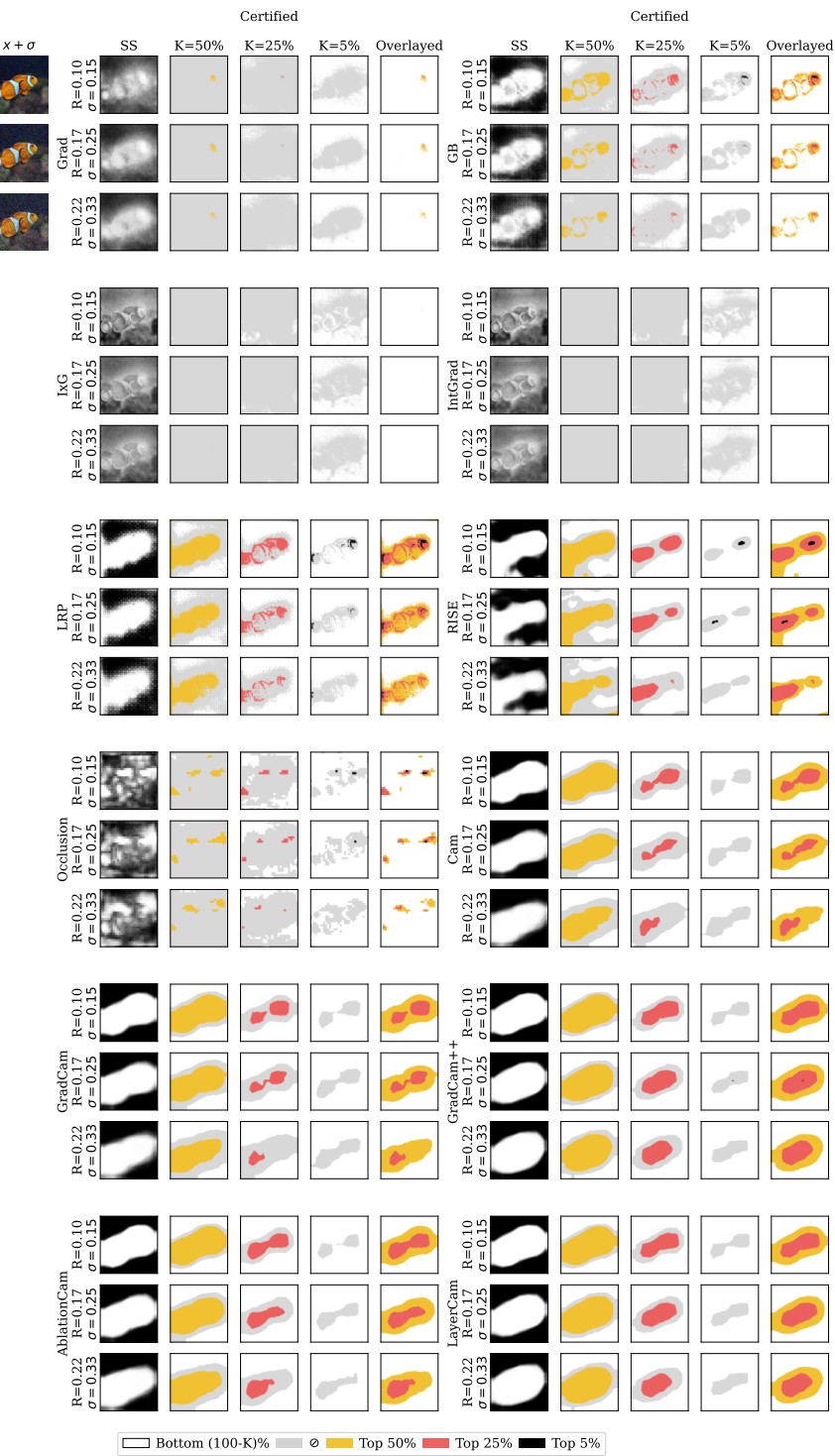

Figure 15: **Example image and its certified attribution maps on ResNet-18 of all methods at different sparsification ($K$) and certified radius $K$ values**. SS (Smoothed Sparsified) is evaluated at $K = 50\%$. The "Overlayed" last column shows the certified top $K\%$ pixels from row-wise certified maps at different $K$ values, with lower $K$ taking precedence for each pixel.

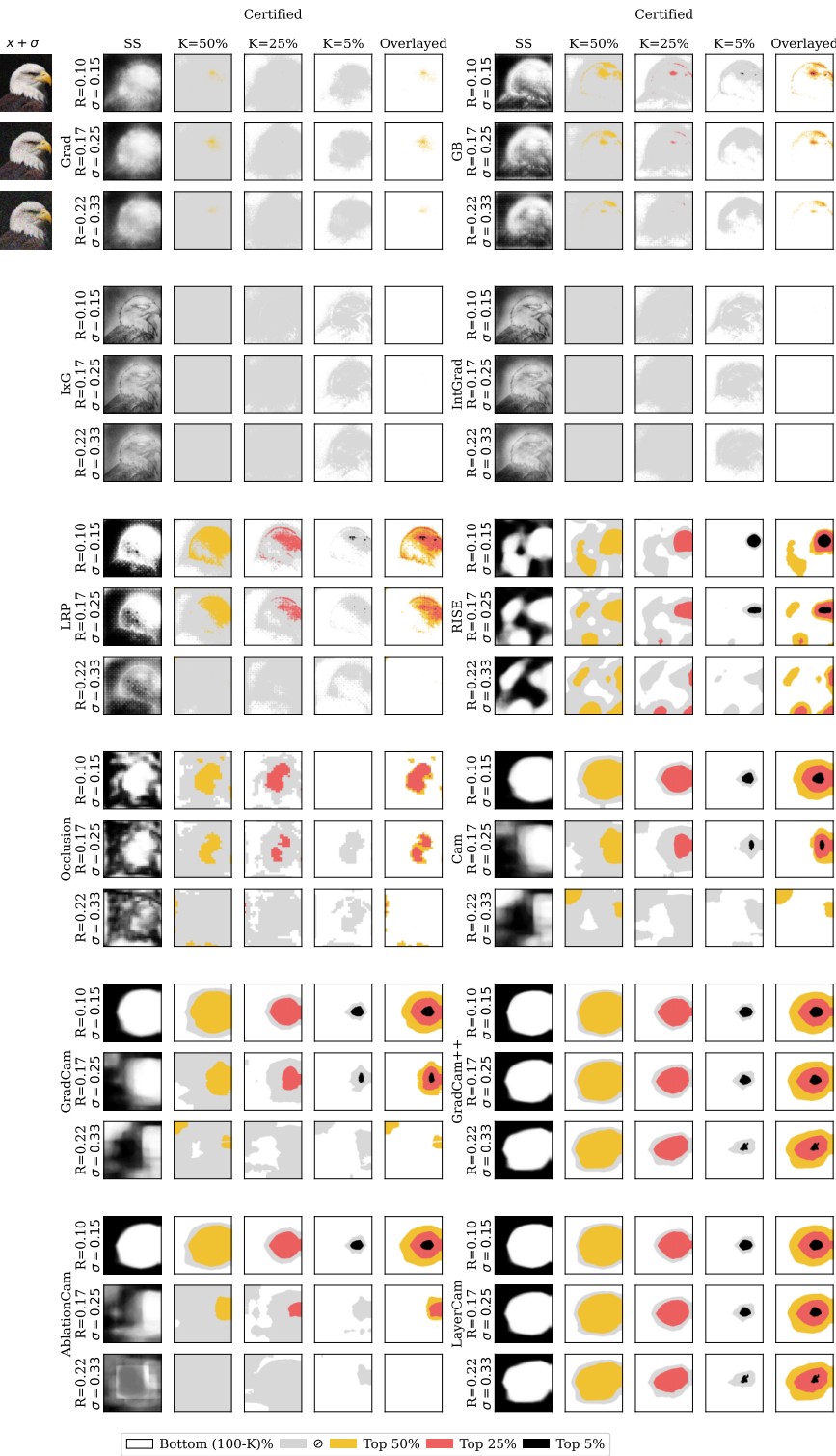

Figure 16: **Example image and its certified attribution maps on ResNet-18 of all methods at different sparsification** ($K$) **and certified radius $K$ values**. SS (Smoothed Sparsified) is evaluated at $K = 50\%$. The "Overlayed" last column shows the certified top $K\%$ pixels from row-wise certified maps at different $K$ values, with lower $K$ taking precedence for each pixel.

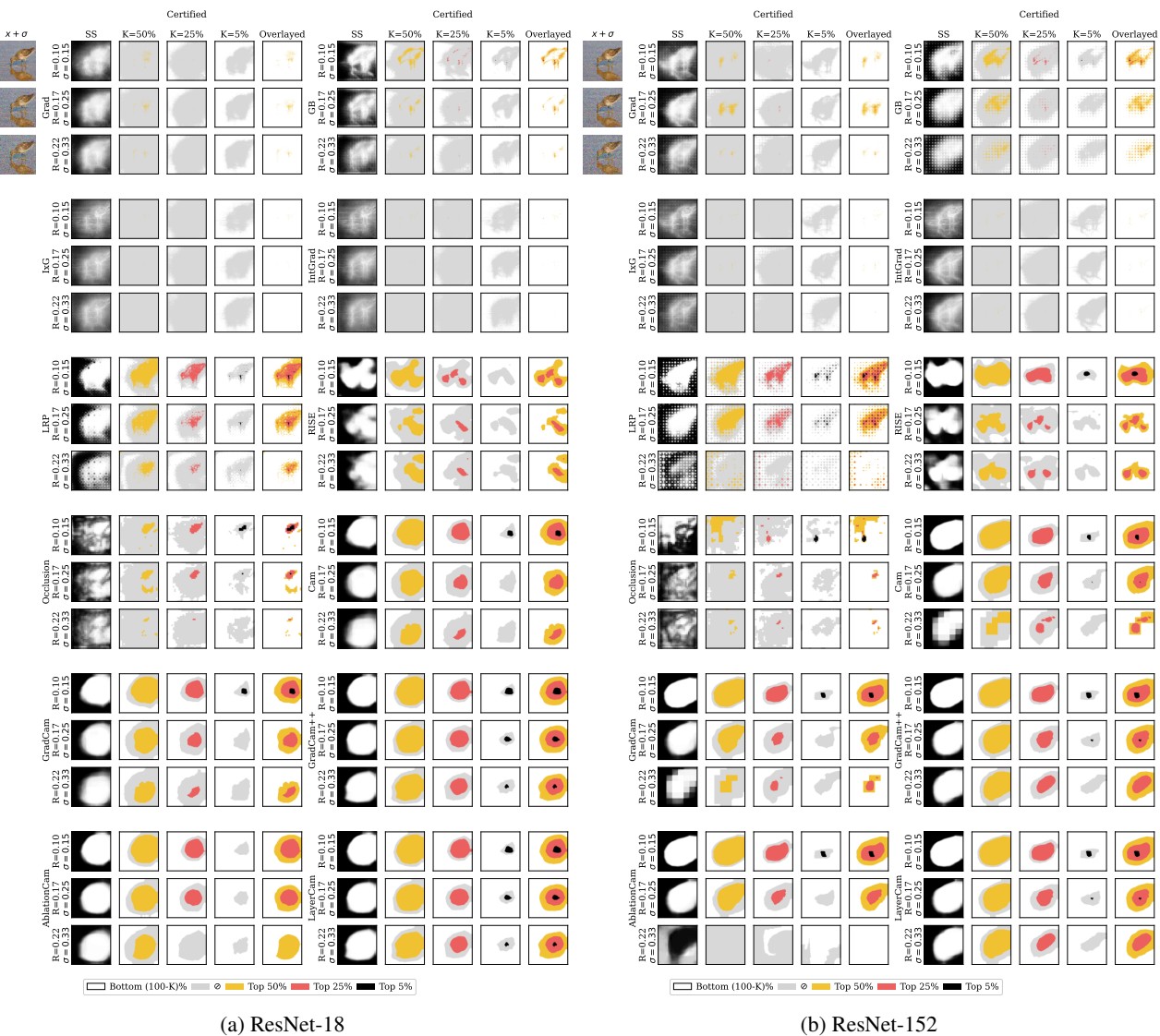

(a) ResNet-18

(b) ResNet-152

Figure 17: **Certified attribution maps on ResNet-18 (a) and ResNet-152 (b) of all methods at different sparsification ($K$) and certified radius $K$ values**. SS (Smoothed Sparsified) is evaluated at $K = 50\%$. The "Overlayed" last column shows the certified top $K\%$ pixels from row-wise certified maps at different $K$ values, with lower $K$ taking precedence for each pixel.

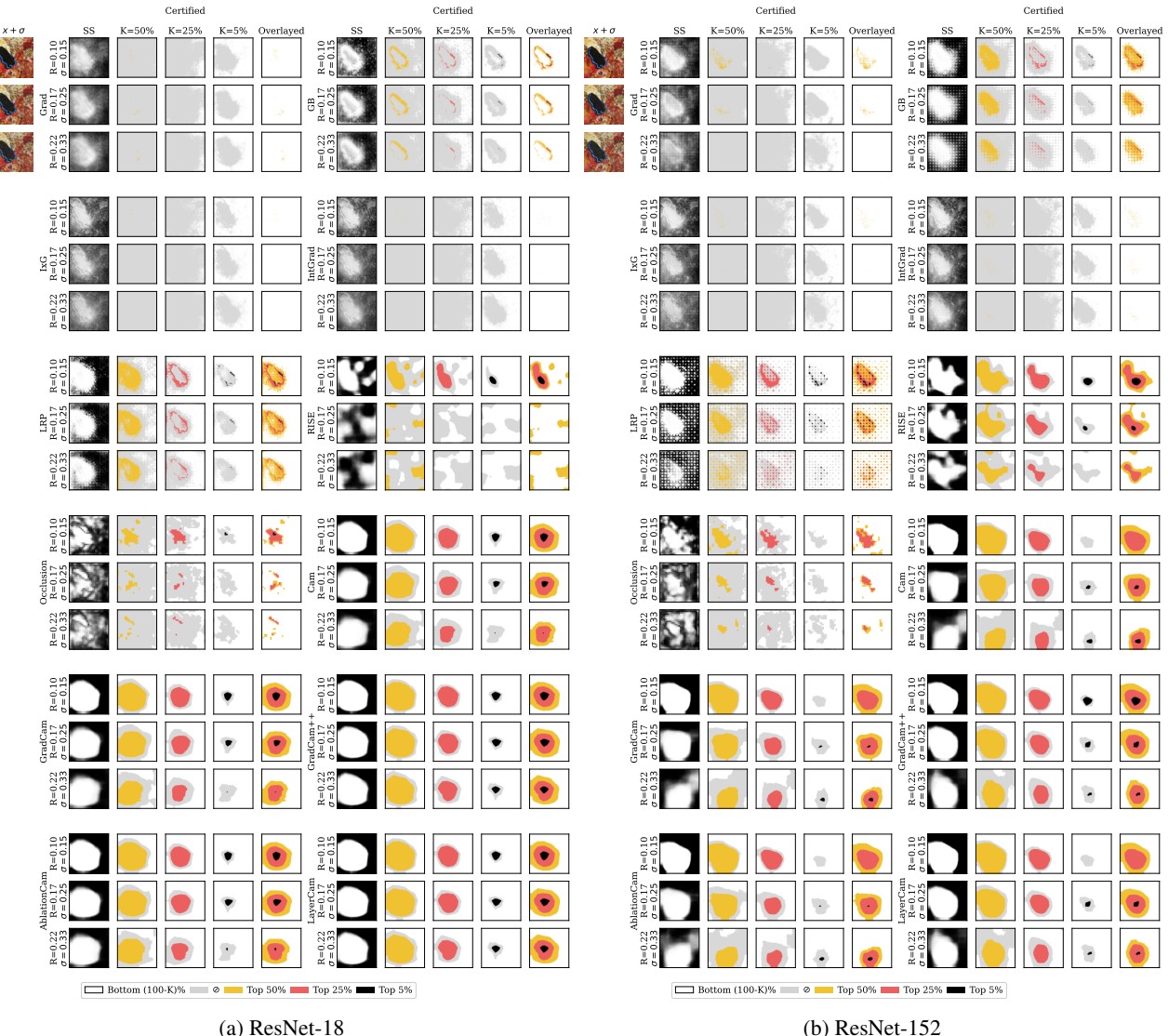

(a) ResNet-18

(b) ResNet-152

Figure 18: **Example image and its certified attribution maps on ResNet-18 (a) and ResNet-152 (b) of all methods at different sparsification ($K$) and certified radius $K$ values**. SS (Smoothed Sparsified) is evaluated at $K = 50\%$. The "Overlayed" last column shows the certified top $K\%$ pixels from row-wise certified maps at different $K$ values, with lower $K$ taking precedence for each pixel.

## F. Certified attribution visuals on ViT-B/16

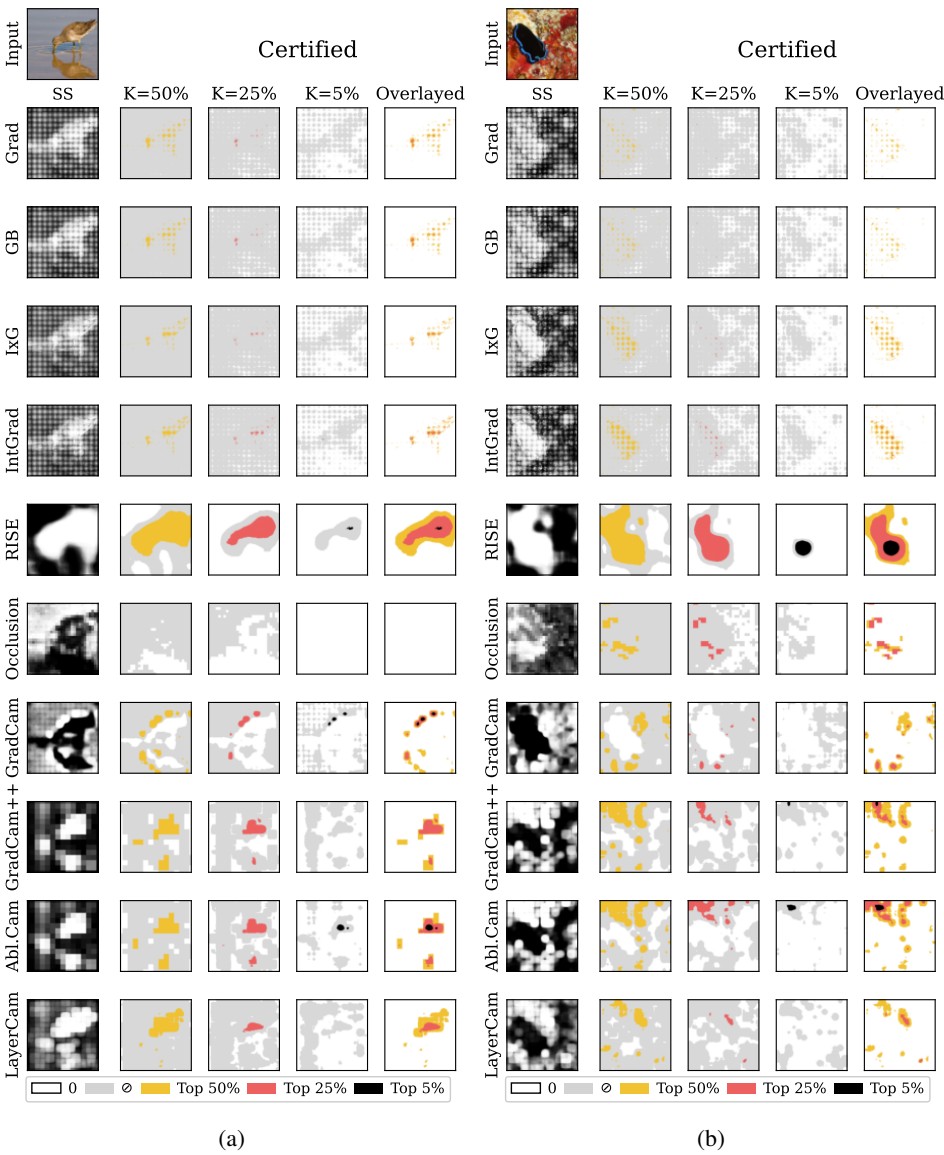

Figure 19: **Certified attribution maps of ViT-B/16 of all methods at different sparsification parameter ($K$) values**. SS (Smoothed Sparsified) is evaluated on $n = 100$ noisy input samples per image and at $K = 50\%$. The "Overlayed" column shows certified top $K\%$ pixels per row, with lower $K$ taking precedence.

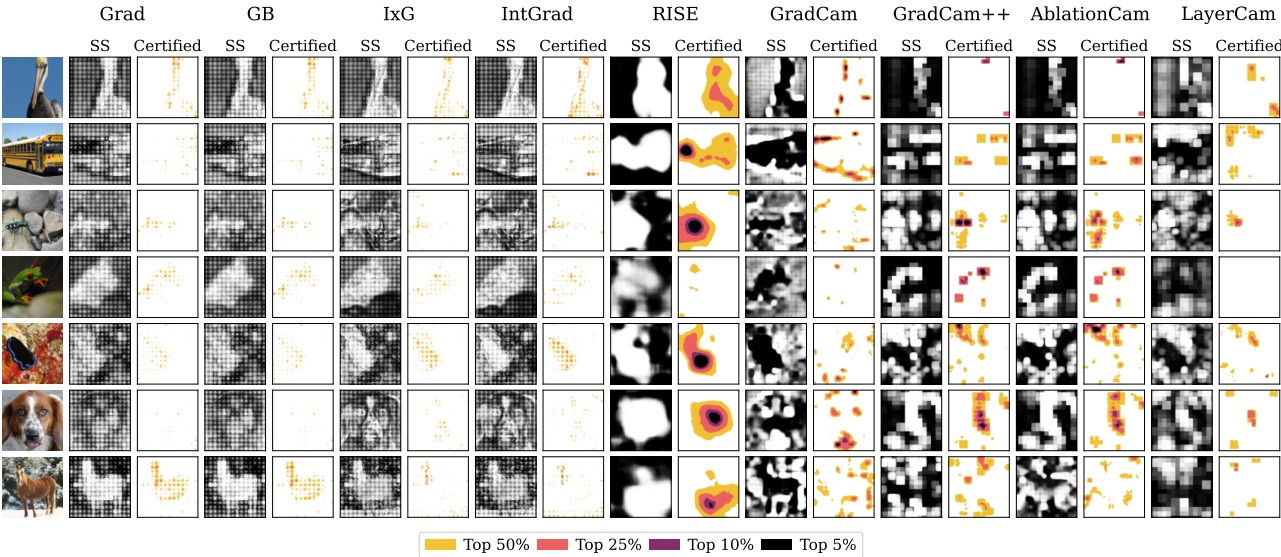

Figure 20: **Overlayed certified attributions on ViT-B/16 at different $K$ values across methods .** SS (Smoothed Sparsified), which refers to the average of the sparsified attributions, is evaluated on $n = 100$ noisy input samples per image and at $K = 50\%$.

# G. Qualitative Evaluation of Certified Localization

In this section, we present qualitative results using the AggAtt layout introduced by Rao et al. (2022), and adapting it for attribution methods evaluated on Certified GridPG at input and final layers. AggAtt (Rao et al., 2022) is a qualitative evaluation method that generates aggregate attribution maps by sorting maps based on their localization scores and grouping them into percentile bins. These bins, with smaller sizes for the top and bottom percentiles, highlight both the best and worst-case performance of attribution methods. This approach provides a comprehensive view of method performance across diverse inputs, emphasizing both consistent trends and distinct failure cases.

In Figures 21 and 22, we show an example from the median position of each AggAtt (Rao et al., 2022) bin for each attribution method at the input and final layers, respectively, evaluated on Certified GridPG at the top-left grid cell using ResNet-18 (He et al., 2016).

At the input layer, **backpropagation-based methods** show relatively better localization in the top-left grid cell. LRP (Bach et al., 2015) shows the best localization across these methods, followed by Grad (Simonyan, 2014). Meanwhile, GB (Springenberg et al., 2014) seems to mainly highlight edges in the input irrespective of the grid cell, and IxG (Shrikumar et al., 2017) and IntGrad (Sundararajan et al., 2017) only have very few pixels certified top $K\%$ pixels within the top-left grid cell. **Activation-based methods** show poor performance by almost abstaining from the entire grid. This aligns with the poor quantitative performance of these methods at the input layer in Figure 6.

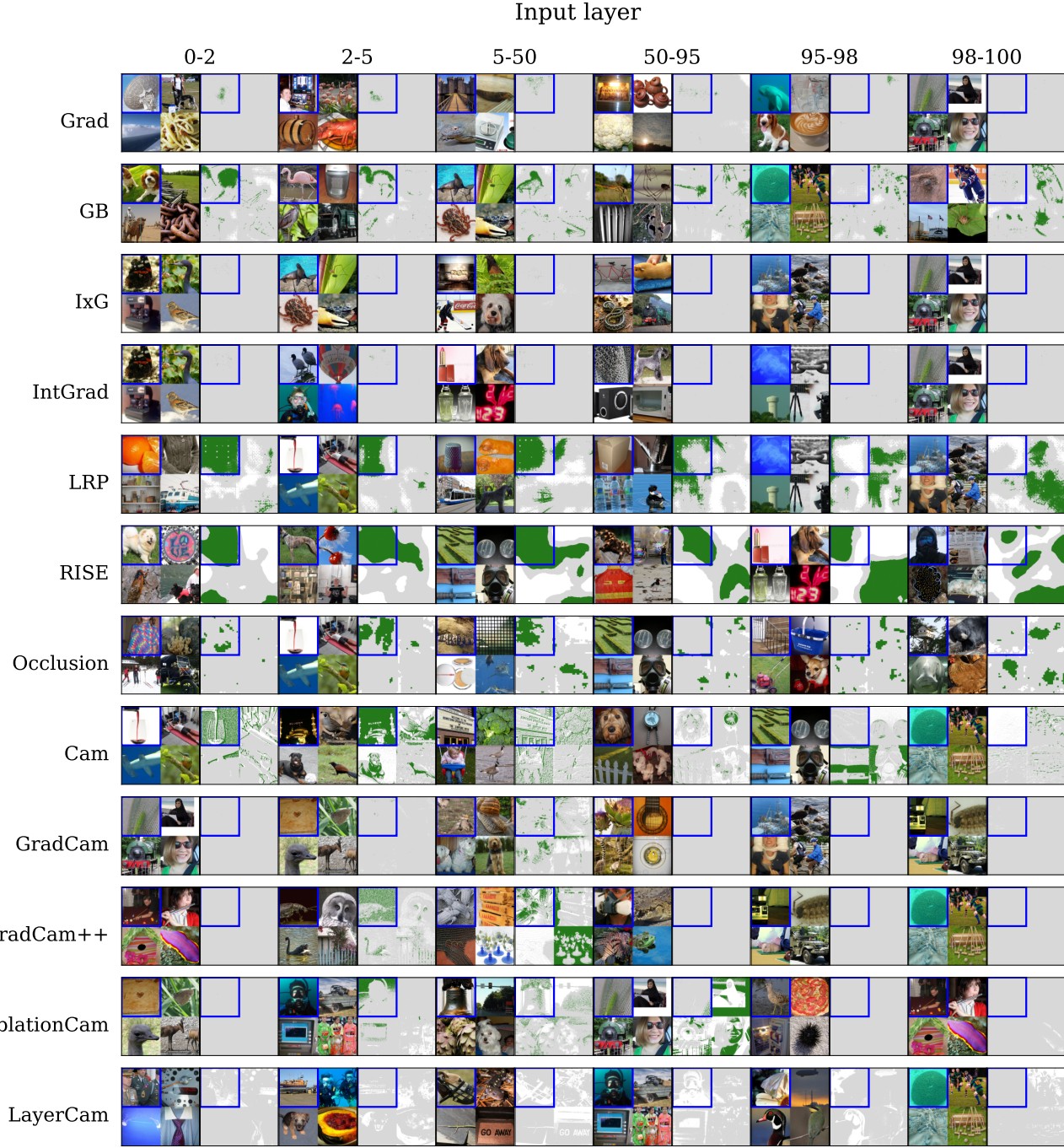

Figure 21: **Examples from each AggAtt for all methods at the input layer using Certified GridPG.** White denotes certified "0", black is certified "1" and gray is abstain ⊘. The percentile bin values are displayed at the top of every column. Default values: $K = 50\%$, $R = 0.10$, $\tau = 0.75$ and $n = 100$.

At the final layer (Figure 22), certified attributions from Grad (Simonyan, 2014) and GB (Springenberg et al., 2014) show a constant pattern where all pixels are certified as bottom $(100 - K)\%$. The localization of the rest of the methods improves considerably compared the input layer in Figure 21, which agrees with the quantitative results from Figure 6. All other methods show good localization, with the best example coming from Occlusion (Zeiler & Fergus, 2014), which achieves near perfect localization at the first $0 - 2$ AggAtt bin.

Final layer

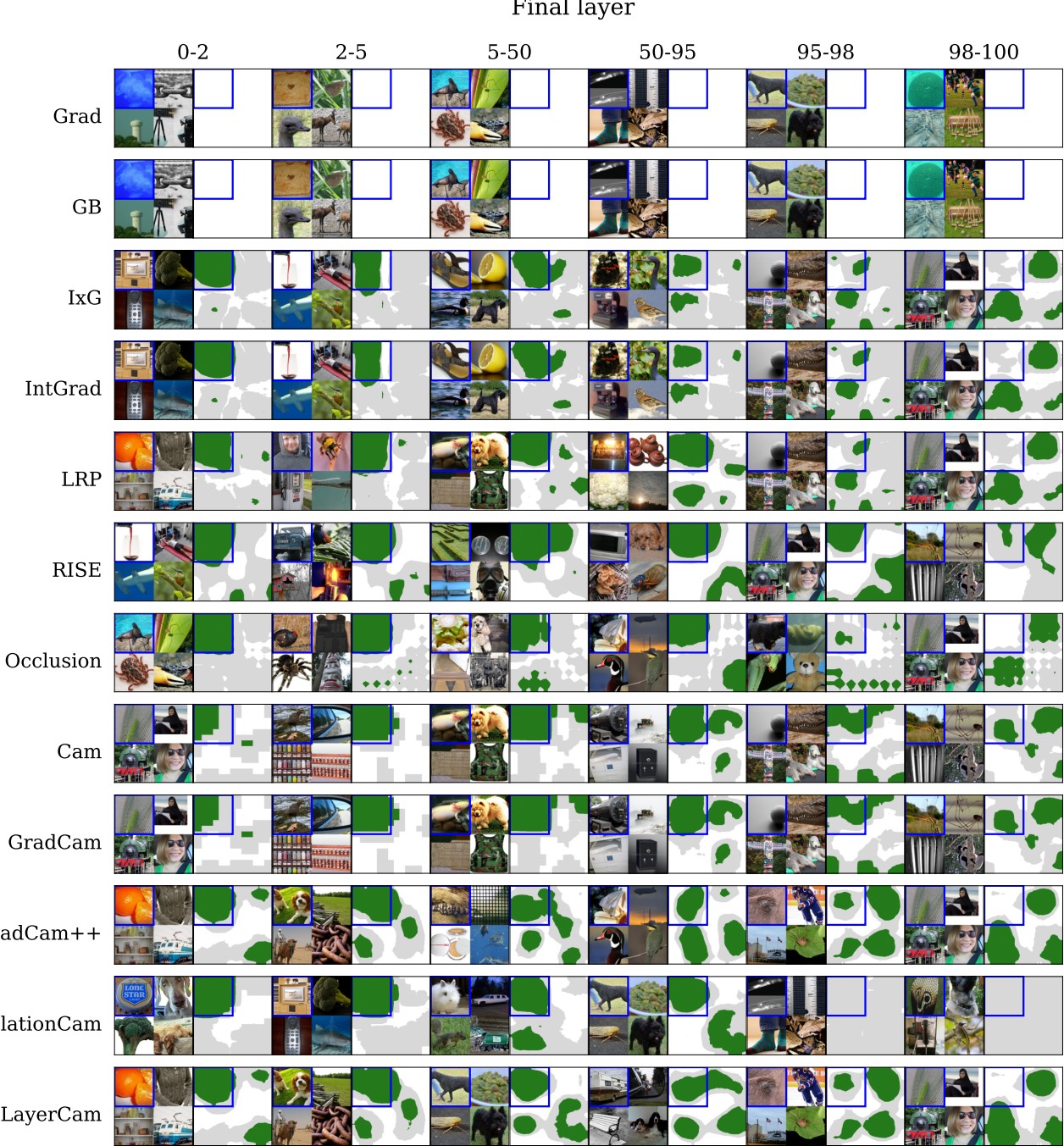

Figure 22: **Examples from each AggAtt for all methods at the final layer using Certified GridPG.** White denotes certified "0", black is certified "1" and gray is abstain $\oslash$. The percentile bin values are displayed at the top of every column. Default values: $K = 50\%$, $R = 0.10$, $\tau = 0.75$ and $n = 100$.

Finally, we show the AggAtt (Rao et al., 2022) bins for all methods on Certified GridPG at both layers at different sparsification values and certified radii in Figure 23. We see that the AggAtt bins in this comprehensive figure reflect the trends in the quantitative results in Figure 5 and Figure 6, as well as the qualitative results in Figure 21 and Figure 22.

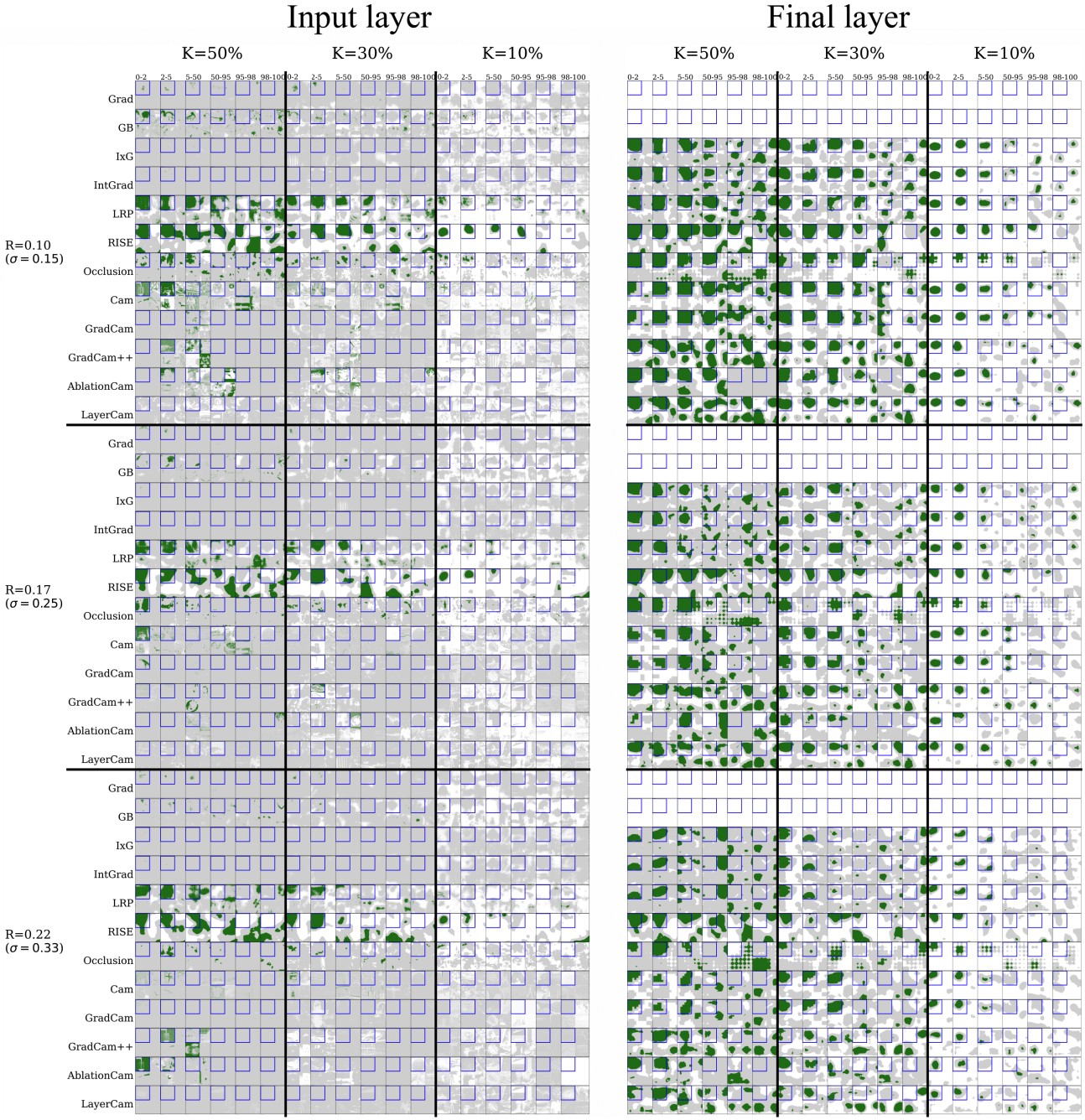

Figure 23: **AggAtt Evaluation on Certified GridPG for all attribution methods at the input and final layers** across sparsification parameters $K$ and certified radii $R$ values.

