# OpenReview forum: "Pixel-level Certified Explanations via Randomized Smoothing"
_ICML.cc/2025/Conference — ICML 2025 poster_

### Official Review · Reviewer_iTN6 · 2025-03-09

**Overall Recommendation:** 4

**Summary:**

The paper introduces a pixel-level certification approach for black-box attribution methods using Randomized Smoothing.

The authors reformulate the pixel-level attribution problem into a segmentation task and provide theoretical guarantees adapted from Fischer et al. (2021).

Extensive experiments demonstrate certified robustness and localization metrics across various attribution methods (3) and provide insights into robustness-localization trade-offs.

### Update the Scores after Rebuttal
After rebuttal and checking the author's additional results + Other's Reviews Comments, I am convinced to raise the score, from weak accept -> Accept :)
Just one tips on Rebuttals, for readability, it is better to keep all the mathematical equation symbols / bold / font consistent (e.g. for Review itgy)!

**Claims And Evidence:**

1) The core theoretical approach heavily relies on prior work by Fischer et al. (2021) (lines 206-219).
2) The paper’s primary novelty appears limited to extending the certification method to pixel-level explanations. However, the paper does not clearly specify which elements of the theoretical framework are genuinely novel beyond the adaptation.
3) Equations 4 and 5 alone are insufficient to justify significant theoretical novelty. Clear articulation of the specific theoretical contributions beyond adaptation is needed.

**Essential References Not Discussed:**

Missing references to recent relevant datasets and benchmarks for explainable AI. There could be even more XAI dataset for testing, only included ImageNet results in the paper.

**Experimental Designs Or Analyses:**

1) The authors extensively evaluate 3 different attribution methods, certification strategies, and the impact of layers and sparsification parameters (Sections 6 and 7), providing robust insights into method performance across several scenarios.


2) However, it does not sufficiently discuss or visually demonstrate the practical benefits of certified pixel-level attributions over standard, widely-used XAI visualization methods such as saliency maps, Grad-CAM, or simpler standalone visualization tools. Specifically, it lacks a clear visual or practical argument about why certified pixel-level explanations significantly enhance real-world usability compared to non-certified alternatives.

**Methods And Evaluation Criteria:**

1) Good Point: the paper features a clear paper structure, covering impact, and ethical considerations.
2) It also provides extensive supplementary materials, including detailed ablation studies on sparsification parameters, radius values, and layer impacts (Section E, Appendix).
3) However, the experimental validation exclusively utilizes the ImageNet dataset with shallow CNN architectures such as ResNet18 and VGG11 (lines 221-227). This narrow focus raises concerns about the generalizability of the proposed method.
4) It is unclear how the certification framework performs on deeper, more contemporary CNN architectures (e.g., ResNet101) or transformer-based models (e.g., Vision Transformers - ViTs).

**Other Comments Or Suggestions:**

No.

**Other Strengths And Weaknesses:**

See comments above.

**Questions For Authors:**

1) Clarify explicitly what new theoretical elements you introduced beyond adapting Fischer et al. (2021) Theorem 3.1.
2) How would your certification framework perform with recent transformer-based models or deeper CNNs like ResNet101? Have such evaluations been considered?
3) How specifically does certification improve real-world interpretability and trust?

**Relation To Broader Scientific Literature:**

Builds upon Fischer et al. (2021), clearly positioning its contributions relative to existing certified segmentation literature. However, further discussion of recent developments (transformer-based methods and deeper CNN architectures) would benefit contextualization.

**Theoretical Claims:**

1) Claims regarding theoretical guarantees at the pixel level (lines 80-82) are appropriately supported through adaptation of existing theorems (Fischer et al., 2021). This theoretical framework is sound and built clearly on Fischer et al. (2021). Strong claims on novelty and contributions (Section 1, lines 116-117) need clearer distinctions from prior theoretical frameworks.
2) Empirical claims regarding the robustness-localization trade-off are supported clearly with extensive experiments (Figures 3, 4, 5).

---

> ### Author Rebuttal · Authors · 2025-04-01
>
> **Note:** We include a general response covering global updates and experiments in our comment to Reviewer **itGy**. Please refer to that comment and the supplementary PDF link there.
>
> ### Q: The method is an extension of Fischer et. al. 2021. Therefore, the theoretical novelty is limited.
>
> While our work indeed builds on Fischer et al. (2021)'s certification for segmentation, adapting it to the certified attribution setting, on the pixel-level, is completely novel and non-trivial. The previous work focuses on segmentation models, but it does not contain (1) **The mathematical formulation** for the pixel-level certified attribution setting, (2) **Certified robustness and localization metrics**, (3) **Rigorous comparison** of the robustness-localization tradeoff of certified attributions.
>
> 1. **Mathematical formulation:** We are the first to formulate pixel-level certified attributions by sparsifying the attribution map (Eq. 4) and casting the result as a segmentation-like function with binary pixel labels (important/unimportant) that are semantically meaningful in the attribution settings.
>
> 2. **Certified robustness and localization metrics:** we introduce two certified metrics to evaluate both the pixel-level certified robustness (%certified) and localization (Certified GridPG) of 12 attribution methods spanning 3 families. Our work is the first to discuss such metrics in a certified attribution setting.
>
> 3. **Rigorous comparison of different attribution methods:** by leveraging our contribution in (1) and (2), we benchmark all methods across five architectures (including CNNs and ViTs), certification radii, and multiple sparsification levels.
>
> In short, our work does not trivially rely on prior work by Fischer et al. (2021) as it includes different novel components that are essential for the certified attribution and evaluation settings that are not obvious apriori. It presents an important data point, and research direction, in the field of certified explainability.
>
> ### Q: Is the method generalizable to deeper CNNs (e.g., ResNet101) or transformer-based model (e.g., ViTs)?
>
> We fully agree with the reviewer that broader evaluation across architectures strengthens our conclusions. In response, we have extended our experimental results beyond ResNet-18 and VGG11 to include three additional architectures: Wide ResNet-50-2, ResNet-152, and a Transformer-based ViT-B/16 (Rebuttal Section 1). This nicely rounds up our analysis of the performance of all attribution methods on many architectures of different design and depths.
>
> ### Q: What are the practical benefits of certified pixel-level attributions against non-certified attributions?
>
> Certified attributions provide users with pixel-level guarantees of robustness under input perturbations, allowing them to assess which regions in an explanation are trustworthy. In contrast, standard attribution methods offer no such reliability; highlighted features may be highly sensitive to small input changes. In safety-critical domains such as medical imaging, this distinction is essential. For example, in tumor detection from X-rays, a region attributed as important but uncertified may be unstable and thus misleading. Our framework enables practitioners to identify and rely only on attributions that are provably robust, or abstain when stability cannot be guaranteed.
>
> Thus, certified pixel-level attributions provide actionable interpretability: they enhance standard attribution maps with robustness guarantees, enabling more informed decisions about when to trust a model’s explanation, and by extension, its prediction. This capability is essential for integrating AI safely into high-stakes domains.
>
> ### Q: Besides certified segmentation, can you discuss more recent certification methods (e.g., using transformer or deeper-CNN)?
>
> The certification setting based on Fischer et al. (2021) is architecture-agnostic and applies to any black-box model.
>
> ### Q: Can you discuss recent datasets and benchmarks for XAI and how they relate to your work?
>
> Our current work focuses on certifying pixel-level robustness of attribution methods, and we evaluate on ImageNet for two reasons: (1) it is the standard benchmark used across prior certified attribution works [2, 3], and (2) it includes complex images where attribution stability is challenging. We leverage the Grid Pointing Game benchmark [4, 1], one of the frameworks for evaluating localization and explanation quality, by introducing Certified GridPG, a certified variant of this benchmark tailored to our setup.
>
> [1] Rao et al. Better understanding differences in attribution methods via systematic evaluations. CVPR, 2022.
>
> [2] Levine et al. Certifiably robust interpretation in deep learning. arXiv, 2019.
>
> [3] Liu et al. Certifiably robust interpretation via renyi differential privacy. AI, 2022.
>
> [4] Böhle et al. Convolutional dynamic alignment networks for interpretable classifications. CVPR, 2021.

---

### Official Review · Reviewer_itGy · 2025-03-11

**Overall Recommendation:** 4

**Summary:**

The authors proposed a method to provide the certified explanation on the pixel level using the Randomized Smoothing approach. While a lot of techniques authors used are not novel, the overall pixel-level approach could provide a more rigorous insight into assessing the attribution methods from the robustness point of view.

## update after rebuttal
Authors updated the experimentation parts including a different (more consistent with other certified segmentation papers) $\sigma$ and more attribution algorithms (like CAM, LRP). Also, they provided more clear qualitative analysis on the certified shapes. I updated the score from WA to Accept.

**Claims And Evidence:**

Claims:
1. a pixel-wise certification approach for attribution methods
2. metrics to compare methods using either the % of certified pixels or Certified GridPG and measuring it for a number of attribution methods (10)
3. rigorous visualization analysis of the certified pixels at different confident levels

Evidences:
1. This claim is supported by the work [1] where Randomized Smoothing (RS) for certification of segmentation is applied
2. Certified analogs of metrics either trivial (%pixels certified) or used from the work [2] making them sound reasonable
3. Multiple figures in the main portion of text as well as Appendices


[1] Fischer, M., Baader, M., and Vechev, M. Scalable certified segmentation via randomized smoothing. In International Conference on Machine Learning (ICML), 2021.
[2] Rao, S., Bo ̈hle, M., and Schiele, B. Better understanding differences in attribution methods via systematic evaluations. Computer Vision and Pattern Recognition (CVPR), 2022.

**Essential References Not Discussed:**

The original seminal work on Class Activation Mappping (CAM) is neither mentioned nor compared - [1]

[1] Zhou, Bolei, et al. "Learning deep features for discriminative localization." Proceedings of the IEEE conference on computer vision and pattern recognition. 2016.

**Experimental Designs Or Analyses:**

There are some important things to consider that I'd like to emphasize:
1. Section 7.2. Why we use radii $R$ = 0.10, 0.17, 0.22? What is the reason behind it? Maybe it is smth used in previous works - need to mention it.
2. Figure 1 uses $K = 40$ while most of others Figures - 2, 6, 7, etc (Appendix A) use $K = 50$. Why not the same $K$?
3. While there are a lot of discussions in Section 7, I haven't noticed any qualitative discussions/conclusions on the different shapes of certified pixel-level explanations (wider area vs concentration on the center etc), and analysis of it w.r.t. how different algorithms behave

**Methods And Evaluation Criteria:**

The metrics proposed to measure the certified attribution look reasonable. A rigorous analysis makes the whole paper sound solid.

**Other Comments Or Suggestions:**

Typos:
* Line 056, "determines" --> "determine"
* Line 638, "certified radius K values" --> R

**Other Strengths And Weaknesses:**

NA

**Questions For Authors:**

Just emphasizing "Experimental Designs Or Analyses" and "Essential References Not Discussed" secitons

**Relation To Broader Scientific Literature:**

Two main areas: Certified Robustness through Randomized Smoothing + Attribution methods.

**Theoretical Claims:**

The only formula-heavy part is borrowed from either work [1] or [2] (section "Claims And Evidence"), so nothing to review.

---

> ### Author Rebuttal · Authors · 2025-04-01
>
> # Response to all reviewers
>
> We thank all reviewers for their thoughtful and constructive feedback. We very much appreciate the diligent and constructive reviews by all reviewers, and believe the additional insights gained in preparing this rebuttal significantly strengthen our work. While we respond to all reviews individually, we would like to draw your attention to the following general points:
>
> ## Supplementary PDF
>
> A supplementary PDF with all figures and analyses is available: https://anonymous.4open.science/r/certified-attributions-E80D/ICML25_Rebuttal___Pixel_level_certified_explanations_via_randomized_smoothing.pdf
>
>
> ## Key Updates
>
> **Three new architectures:** Added Wide ResNet-50-2, ResNet-152, and ViT-B/16 (**WAuF**, **iTN6**)
>
> **Faithfulness metric:** Deletion-based faithfulness analysis of certified attributions (**S6h4**)
>
> **Two new attribution methods:** Added CAM and LRP (**itGy**)
>
> **Qualitative visuals:** More certified maps for shape and Certified GridPG analysis (**WAuF**, **itGy**)
>
> ## Conclusions
>
> - Our certification setup is general enough to scale across **five different architectures** and **twelve attribution methods**.
>
> - **Certified robustness, certified localization and empirical faithfulness are not mutually exclusive**, and some methods strike a balance between the three (e.g., LRP), hence, providing faithful and trustworthy attributions, essential for safety-critical domains.
>
> Sincerely,
>
> The authors
>
> # Response to Reviewer itGy
>
> ### Q: Why do you use radii $R= 0.10\, 0.17\, 0.22$?
>
> We follow the common practice from [1] and [2] for Randomizerd Smoothing for segmentation for such values of the radius. The certified radius $R\coloneqq \frac{\sigma}{2} \Phi^{-1}(\tau)$ depends on the noise level $\sigma$ and the top class probability $\tau$ (Section 4.2). We fix $\tau=0.75$ and vary $\sigma \in \{0.15, 0.25, 0.33\}$ to obtain different certified radii. These values ensure the added noise remains visually imperceptible, while enabling a controlled evaluation of robustness across a range of certification strengths.
>
> ### Q: Why does Figure 1 use K=40, while the rest use K=50?
>
> We ran all the analysis with the same $K=40$ and again double-checked this in our codebase. It turned out to be just a typo, Figure 1 indeed uses $K=50$ like the remaining ones. Thank you for pointing this out!
>
> ### Q: Can you provide qualitative discussions on shapes of certified pixel-level explanations across methods?
>
> Certainly! We extend the qualitative comparison of certified attribution shapes in Section 7.1 and further detailed in App. A & E.
>
> We observe that input-layer methods (e.g., Backpropagation and perturbation) produce fine-grained certified regions, often concentrated on small parts of the object (e.g., GB highlighting eyes in Figure 1). In our rebuttal experiments, LRP nicely highlights the object boundaries and its parts with varying importance levels in Rebuttal Figures 10 & 11. In contrast, activation-based methods evaluated at the final layer produce larger and coarser certified regions. More specifically:
>
> - **Backpropagation-based methods** are highly sensitive to perturbations and often abstain, but when certified top $K$% pixels are present, they highlight sharp localized features. LRP seems to provide the sharpest certified attributions.
>
> - **Perturbation-based methods’** coarseness of highlighted regions is influenced by the mask size and generally yield broader certified top $K$% regions than backpropagation-based methods. They also exhibit higher robustness.
>
> - **Activation-based methods** produce lower-dimensional attributions that, when rescaled to the input size, yield coarser certified top $K$% regions. Our framework nicely controls the granularity of such regions via the sparsification parameter $K$, lower values concentrate the certified top $K$% pixels on smaller more important areas (Figure 2).
>
> ### Q: Can you mention or compare the CAM method in your evaluation pipeline?
>
> Thank you for this suggestion. We now include results on not only ten attribution methods, but also two additional ones: Class Activation Mapping (CAM) [3] and Layer-wise Relevance Propagation (LRP) [4] in Rebuttal Section 3. We notice that Cam highlights coarse certified top $K%$ regions similar to other activation methods, while LRP produces high-quality certified attributions that focus on fine-grained details in the objects in Rebuttal Figures 10 & 11.
>
> ### Q: Should L056 and L638 be changed?
> Yes, thanks for pointing the typos out!
>
>
> [1] Fischer et. al. Scalable certified segmentation via randomized smoothing. ICML, 2021.
>
> [2] Anani et. al. Adaptive hierarchical certification for segmentation using randomized smoothing. ICML, 2024.
>
> [3] Zhou et al. Learning deep features for discriminative localization. CVPR, 2016.
>
> [4] Bach et al. On pixel-wise explanations for non-linear classifier decisions by layer-wise relevance propagation. PloS one, 2015.

---

> > ### Comment · Reviewer_itGy · 2025-04-02
> >
> > Thank authors for replying the comments and especially comparing with CAM/LRP and providing more shape-level analysis on certification.
> >
> > I want to follow up with the following:
> > * "We follow the common practice ... for Randomizerd Smoothing for segmentation for such values of the radius. ... We fix $\tau=0.75$ and vary $\sigma \in {0.15, 0.25, 0.33}$ to obtain different certified radii"
> >
> > Actually, in referred papers the following set of sigmas used: 0.25, 0.33, and 0.5 - that makes their task harder (you are using 0.15 instead of 0.5). Could you please comment on it / measure for 0.5?

---

> > > ### Author Response · Authors · 2025-04-03
> > >
> > > We thank the reviewer for their thoughtful follow-up and for acknowledging our comparisons with CAM, LRP, and the inclusion of shape-level analysis.
> > >
> > > **Regarding the exclusion of the highest noise level $\sigma = 0.5$ in our experiments:**
> > >
> > > We appreciate the reviewer’s observation that prior works [1, 2] have included one higher noise level ($\sigma=0.5$) in their experiments.
> > >
> > > **First**, for consistency, we provide additional experiments at $\sigma=0.5$ in Section 5 (see
> > > https://anonymous.4open.science/r/rebuttal-comments-4256/ICML25_Rebuttal_V2_Pixel_level_certified_explanations_via_randomized_smoothing.pdf).
> > >
> > > **Second**, in our experiments, we deliberately limited the noise level to $\sigma \leq 0.33$ for the following reasons:
> > >
> > > 1. Most methods already yield either near-zero %certified or near-random Certified GridPG at $\sigma = 0.33$ (see Rebuttal Figure 12). Thus, further increasing $\sigma$ offers limited practical value.
> > >
> > > 2. Attribution is more sensitive than segmentation models: we observe that the abstention rate (i.e., 1-%certified)  is substantially higher in certified attributions than in segmentation at the same radii. For example, in Table 1 of [1], the abstention rate (%$\oslash$) for the segmentation model reaches only $20$% at $\sigma=0.33$ ($R=0.22$), whereas in certified attributions, it can reach 100% in some methods (Rebuttal Figure 12). Therefore, the two tasks are not directly comparable.
> > >
> > > In Rebuttal Figure 12  _(bottom row)_, we show that at the largest radius $R = 0.34$ ($\sigma=0.5$), all methods yield near-random Certified GridPG scores, indicating a loss of meaningful localization. Although some methods retain non-zero %certified values _(top row)_, these correspond to semantically uninformative regions, indicating robustness without interpretability. In Figure 13, we actually visualize the certified attributions at $R=0.34$, where the outputs are largely uninformative (with few Grad-CAM++ exceptions), as expected under such high noise.
> > >
> > > In conclusion, while certification at higher radii does indeed provide stronger theoretical guarantees, the empirical results suggest that attribution quality substantially degrades under such high noise.

---

### Official Review · Reviewer_WAuF · 2025-03-12

**Overall Recommendation:** 1

**Summary:**

The article proposes a certification method that reformulates the attribution task as a segmentation problem and uses the randomized smoothing technique to ensure the pixel-level robustness of black-box attribution methods. Meanwhile, two metrics, "percentage of certified pixels" and "Certified Grid Pointing Game", are introduced to evaluate attribution methods from the aspects of robustness and localization ability. The experiments use the ImageNet dataset and the ResNet18 model to conduct research on ten attribution methods from three categories.

**Claims And Evidence:**

Yes

**Essential References Not Discussed:**

No

**Experimental Designs Or Analyses:**

Yes

**Methods And Evaluation Criteria:**

It is applicable, but the number of experimental images and the number of selected models are too limited.

**Other Comments Or Suggestions:**

No

**Other Strengths And Weaknesses:**

Strength
The paper proposes a method that reformulates the attribution task as a segmentation problem and uses the randomized smoothing technique to achieve pixel - level robustness certification. This breaks through the limitation of only providing image - level bounds in the past, opening up a new direction for the research on the interpretability of deep learning. It can more accurately evaluate the robustness of pixel importance, enhance the reliability of model interpretations, and is of great significance for applications in safety - critical fields.

Weaknesses
The experiments only use the ImageNet dataset and models like ResNet18 and VGG11. The selection of the dataset and models is relatively limited, which may not comprehensively reflect the performance of the method under different data distributions and model architectures, thus restricting the generality of the research results.

The evaluation of attribution methods mainly relies on two indicators, "percentage of certified pixels" and "Certified Grid Pointing Game". This may not cover all important aspects of the performance of attribution methods, ignoring other factors that may affect the effectiveness and reliability of the methods, resulting in an incomplete evaluation.

The Monte Carlo sampling method is used in the certification process. Due to the large number of samples, the computational cost is high. This not only limits the application efficiency of the method on large - scale data and complex models but also may affect its real - time performance and practicality in actual scenarios.

**Questions For Authors:**

The certification method proposed in the paper is applied to the ImageNet dataset in the experiment. When dealing with different types of image datasets, such as medical images and satellite images with special features and distributions, can it still maintain good performance and robustness? If applied to these datasets, does the method need to be adjusted or improved?

Are the two metrics, "percentage of certified pixels" and "Certified Grid Pointing Game", comprehensive enough to measure the performance of attribution methods? Are there other important aspects that are not covered by these two metrics? For example, in complex scenarios, is there a lack of evaluation of the localization ability for small targets?

The paper mentions that the high computational cost of sampling in the certification process may limit its practical application. In future research, is there a plan to explore more efficient sampling strategies or computational methods to reduce the computational cost and improve the scalability of the method?

The experiments are mainly based on ResNet18 and VGG11 models. For models with other different architectures (such as Transformer - based architectures), is this certification method equally applicable? Different architectures vary in feature extraction and representation. How will this affect the certification results?

**Relation To Broader Scientific Literature:**

In the research of machine learning interpretability, post - hoc attribution methods can indicate the impact of image pixels on classifier predictions. However, they are not robust to minor changes in the input. Previous studies have used regularization techniques or improved sampling methods to enhance robustness, but these lack theoretical guarantees. The pixel - level certification method proposed in this paper addresses this shortcoming. By reformulating the attribution task as a segmentation problem and leveraging the randomized smoothing technique, it theoretically ensures the pixel - level robustness of black - box attribution methods.

**Theoretical Claims:**

Yes

---

> ### Author Rebuttal · Authors · 2025-04-01
>
> **Note:** We include a general response covering global updates and experiments in our comment to Reviewer **itGy**. Please refer to that comment and the supplementary PDF link there.
>
> ### We thank the reviewer for recognizing the significance of our certification framework, particularly the value of providing pixel-level robustness guarantees and its relevance in safety-critical applications.
>
>
> ### Q: Should the method be evaluated on different architectures?
>
> We agree that broader evaluation across architectures strengthens our conclusions. We extend our experimental results to 3 additional architectures: Wide ResNet-50-2, ResNet-152, and a Transformer-based ViT-B/16. These include deeper CNNs and a transformer, confirming the generality of our method (Rebuttal Section 1).
>
> ### Q: Can you increase the readability and cleanliness of the code?
>
> We appreciate this suggestion. We are restructuring the codebase to include a clear execution guide, ensuring full reproducibility. This will be available in the camera-ready version.
>
> ### Q: Should the method be evaluated under different data distributions?
>
> We used ImageNet to align with prior certified attribution work. We agree that it would be advantageous to extend our evaluation to a different data distribution. Due to limited computational resources during the rebuttal, we prioritized architectural diversity. That said, our method applies directly to other data domains, and we will include such evaluations in the final version.
>
> ### Q: Should you cover all important aspects of the performance of attribution methods (e.g., other than %certified and localization)?
>
> We appreciate the concern and would like to clarify a key point: the goal of our work is **not** to introduce new, but **certify the robustness of existing attribution methods**, which means their original performance on standard metrics *stays unchanged*. Our certification provides context for the end user that allows them to asses which explanations (pixel-wise attributions) to **trust**. To this end, our proposed two certified metrics answer the questions:
>
> 1. Does the explanation remain stable under perturbations? (%certified)
>
> 2. Do the certifiably robust pixels point to the correct region? (Certified GridPG)
>
> In summary, our work does not aim to redefine attribution performance metrics, but to provide a certified framework that augments existing methods with trust guarantees, allowing practitioners to assess the trustworthiness of attributions. Our evaluation framework is designed to complement, not replace, standard evaluation.
>
> ### Q: Can you comment on the computational cost of the method, and its real-time performance?
>
> Certainly! We agree with the reviewer that the certification process incurs a computational cost due to the Monte Carlo sampling. We use $n=100$ samples per image in our experiments, which suffices to obtain statistically sound guarantees at a reasonable cost. In fact, there is actually seminal work on sampling with noise just to sharpen gradient-based attribution maps, such as SmoothGrad [2], which is shown to be most faithful by a benchmark for interpretability [3]. While, like our work, not designed for real-time deployment, our method is suitable for high-stakes offline settings such as medical diagnostics, legal forensics, and model auditing, where certifiable robustness is essential.
>
> ### Q: Does the method need to be adjusted or improved when dealing with different dataset types?
>
> Thank you for raising this question, which highlights the practical applicability of our method. Our certification pipeline is model and dataset-agnostic. It treats the attribution function $h(x)$ as a black-box and operates without assumptions on the model or input distribution. Thus, it applies to any domain where attribution is defined.
>
> ### Q: Does Certified GridPG cover the localization of small targets?
>
> Yes, **Certified GridPG** is explicitly designed to be **invariant** to object **size**. It computes the fraction of certified top $K$% pixels located in the correct subimage within a constructed grid (e.g., $2\times 2$), regardless of the object’s size (Eq. 8). This formulation extends [1]. Rebuttal Figure 11 and App. E (Figures 12-14), show various qualitative results across layers, certification radii, and sparsification levels.
>
> ### Q: Is the certification equally applicable to transformer-based architectures?
>
> Certainly! Our approach does not rely on any architectural properties. We have validated this by including ViT-B/16 in our extended experiments, alongside deeper CNNs (Rebuttal Section 1).
>
> [1] Böhle et al. Convolutional dynamic alignment networks for interpretable classifications. CVPR, 2021
>
> [2] Smilkov et al. Smoothgrad: removing noise by adding noise. arXiv, 2017
>
> [3] Hooker et al. A benchmark for interpretability methods in deep neural networks. NeurIPS, 2018

---

### Official Review · Reviewer_s6h4 · 2025-03-16

**Overall Recommendation:** 3

**Summary:**

The paper’s primary contribution is a pixel-level certification framework for attribution methods on deep classifiers, backed by a robust evaluation paradigm and demonstrated through high-quality visuals and quantitative comparisons.

**Claims And Evidence:**

The claims in this paper are well-supported and authors have provided empirical results demonstrating the validity of the claims.

**Essential References Not Discussed:**

There is an import stream of work in the relation between model robustness and attribution robustness that are not discussed in this work. Example references are listed in Methods And Evaluation Criteria.

**Experimental Designs Or Analyses:**

I think experiments are useful to support the paper's goal, which is actually problematic. As a result, some important evaluations are missing, such as attribution faithfulness. See my comments in Methods And Evaluation Criteria.

**Methods And Evaluation Criteria:**

This is the biggest concern I have regarding this paper. Although the paper's method and results are supporting each other, I am quite confused of what problem this is paper is trying to address, and if this work is still creating an "explanation".

First, this paper starts with a claim that is not well-accepted in the abstract:

> Certified attribution methods should prove that pixel-level importance values are robust.

I believe this claim is not well-received. Can the authors provide a step-by-step reasoning here. Normally, we provide a certification to a function to check if its output is consistent within a support set (e.g. among adversarial inputs). I am not clear what do we mean by creating certifiable robust explanations, the reasons are as follow:

For the case of input attribution, the explanation means a subset of input features that are most important to the model. For an attribution to be useful, it must be faithful to the model in the first place [1, 2, 3]. There are many definitions of faithfulness, but in the simplest way at least we want the input attribution to be as sensitive to the input as the model. That is, if the model changes the prediction with some adversarial examples, the input attribution to the benign and the adversarial input should be different. As a result, we observe that attribution robustness naturally come with model robustness [3, 4, 5].

However, the goal of this paper is trying to make the attribution robust, without discussing the robustness of the model itself. Thus, my read is the more robust an attribution is under your certification, it is LESS likely to be a *faithful* attribution.

Can the authors provide some analysis and discussions on whether the certified attribution in this work still accurately capture the model's behavior? You can evaluate the faithfulness scores such as [2].


[1] Sundararajan, M., Taly, A., and Yan, Q. Axiomatic attribution for deep networks. In International Conference on
Machine Learning (ICML), 2017.

[2] Yeh, Chih-Kuan, et al. "On the (in) fidelity and sensitivity of explanations." Advances in neural information processing systems 32 (2019).

[3] Wang, Zifan, Matt Fredrikson, and Anupam Datta. "Robust models are more interpretable because attributions look normal." arXiv preprint arXiv:2103.11257 (2021)

[4] Prasad Chalasani, Jiefeng Chen, Amrita Roy Chowdhury, Xi Wu, and Somesh Jha. Concise explanations of neural networks using adversarial training. In International Conference on Machine Learning, pp. 1383–1391. PMLR, 2020.

[5] Etmann, C., Lunz, S., Maass, P., and Schoenlieb, C. On the connection between adversarial robustness and saliency map interpretability. In Proceedings of the 36th International Conference on Machine Learning, 2019.

**Other Comments Or Suggestions:**

N/A

**Other Strengths And Weaknesses:**

N/A

**Questions For Authors:**

Please see Methods And Evaluation Criteria

**Relation To Broader Scientific Literature:**

This paper is based on randomized smoothing, a technique used to provide probabilistic guarantee of a function. The authors tried to use this method to provide a guarantee of input attribution method, an approach used in explaining a deep model's local decision by identifying a subset of input features that are more relevant than the whole input.

**Theoretical Claims:**

More analytical results are using conclusions or some direct derivatives from randomized smoothing, so they are sound.

---

> ### Author Rebuttal · Authors · 2025-04-01
>
> **Note:** We include a general response covering global updates and experiments in our comment to Reviewer **itGy**. Please refer to that comment and the supplementary PDF link there.
>
> We thank the reviewer for raising very fundamental questions regarding the motivation of our work. We clarify that our goal is **not to make attributions robust**, but to **certify their robustness** at the pixel level under input perturbations. This distinction appears to have been misunderstood. Other reviewers recognized our contribution: **WAuF** noted that our method opens “a new direction” in interpretability and **iTN6** highlighted our “robust insights into method performance”.
>
> ### Q: What is the problem this paper is trying to address?
>
> We address the lack of pixel-level robustness guarantees in post hoc attribution methods. As discussed in (Section 2, Robustness issues), small input perturbations can significantly alter attribution maps even when the predictions remain unchanged, undermining trust in explanations, especially in high-stakes domains. Prior work provides only image-level bounds (Section 2, Certified explanations), which lack interpretability (Section 1, L023-L037 Right). We bridge this gap by introducing a certification setting that provides pixel-level guarantees, along with certified metrics for robustness and localization.
>
> ### Q: Step-by-step reasoning for “pixel-level robustness should be proven”.
>
> 1. For a model to be trustworthy, its explanations must also be trustworthy.
> 2. Attribution methods are non-robust under small perturbations.
> 3. Existing certified attribution work focuses only non-interpretable image-level numeric bounds.
> 4. Pixel-level guarantees are essential to identify trustworthy regions in explanations. In a biomedical setting such as tumor detection from X-rays, a doctor can directly see which pixels the model used to predict cancer. For example, if a model highlights a shaded region as important, but our certification flags it as non-robust, a clinician can be alerted to treat the attribution with caution, potentially influencing their reliance on the model’s prediction.
> 5. Thus, the claim is justified to ensure trust in explanations.
>
> ### Q: Does the method consider model robustness?
>
> We agree that model robustness influences attribution robustness [3–5]. However, our goal is to certify attribution robustness for a fixed model, without enforcing or assuming model robustness. **This distinction is crucial**: our certification setting is **attribution-centric**.
>
> To ensure attributions reflect correct class evidence, we select inputs with prediction confidence $\geq 0.99$ (Section 6, L224-L227, Right) as in [6], ensuring certified top $K$% pixels reflect correct class evidence.
>
> ### Q: What do you mean by certifiably robust explanations?
>
> Attribution maps where each pixel's certified label (important/unimportant) is guaranteed to remain unchanged under input perturbations within an $\ell_2$-ball around the input (Eq. 5 & Section 3.2). Unstable pixels are abstained from.
>
> ### Claim: ”The goal of this paper is trying to make the attribution robust.”
>
> **Clarification:** our goal is to **certify, not improve, attribution robustness**. If a method is not robust, it yields high abstentions and a low %certified scores.
>
> ### Q: Does explanation robustness align with faithfulness?
>
> We agree that attribution robustness doesn’t always reflect faithfulness. **High robustness** may be (i) **Informative**, highlighting class-relevant features (i.e., faithful) or (ii) **Trivial**, by producing constant maps. To disentangle these cases, we introduced Certified GridPG (Eq. 8) to jointly assess robustness and localization, which is actually a very effective sanity check. In Rebuttal Fig. 5, RISE and LRP are both robust and localize well, while methods like Grad and GB are robust at the final layer but fail to localize, due to the constant grid-like patterns they produce on that layer.
>
> ### Q: Can you evaluate faithfulness using existing metrics?
>
> Certainly! We thank the reviewer for this valuable suggestion which we believe makes our submission stronger. We include a deletion-based faithfulness analysis in Rebuttal Sec 2, where we progressively remove the certified top K% pixels (lowest to highest) in the input and measure the ground truth class confidence:
>
> 1. At the input layer, LRP and RISE cause the steepest confidence drop in the first removal step, indicating high faithfulness.
> 2. At the final layer, most methods cause a prediction flip after the first deletion step, showing higher faithfulness to the input layer.
>
> Our analysis shows that **high certified robustness and faithfulness co-exist** (e.g., LRP) . Our framework allows the joint evaluation to identify attribution methods that are robust, faithful, and well-localized.
>
> Further analysis in Rebuttal Sec 2.
>
> [6] Rao et al. Better understanding differences in attribution methods via systematic evaluations. CVPR, 2022

---

> > ### Comment · Reviewer_s6h4 · 2025-04-06
> >
> > Thank you for the comprehensive feedback. There might be misunderstanding from my read to this paper, but it might be the case the presentation was not clear enough. Thanks for adding the ablation study. Please update the manuscript to include the discussion here and perhaps link the faithfulness experiments. I will increase my score to 3

---

> > > ### Author Response · Authors · 2025-04-06
> > >
> > > We sincerely thank the reviewer for their thoughtful reconsideration and for raising the score. We are glad the clarifications and faithfulness analysis addressed the concerns.
> > >
> > > We will revise the manuscript to (i) clarify the motivation and formal problem setting, and (ii) include the deletion-based faithfulness experiments into Section 7, explicitly linking them to the certified robustness and localization results.
> > >
> > > We believe this provides a more complete and coherent evaluation.
> > >
> > > Thanks,
> > >
> > > The authors

---

### Decision · Program_Chairs · 2025-05-01

**Decision:**

Accept (poster)

**Comment:**

This paper introduces a pixel-level certification framework for attribution methods, building on randomized smoothing techniques to provide robustness guarantees for saliency-based explanations. The approach is well-executed and includes comprehensive experimental validation, demonstrating its applicability across a range of attribution methods and visualizing the trade-offs between robustness and localization.
Most reviewers initially had concerns but were convinced by the rebuttal. They appreciate the thorough experimental analysis and find the method solid. Reviewer WAuF criticises the limited scope (few models/datasets), the reliance on only two indicators and potential scalability issues.
Taken together, however, the positive opinions about the paper, e.g., it "opens a new direction in interpretability", outweighs the negative feedback.